# A Convergent Gradient Descent Algorithm for Rank Minimization and Semidefinite Programming from Random Linear Measurements

**Qinqing Zheng**
University of Chicago
qinqing@cs.uchicago.edu

**John Lafferty**
University of Chicago
lafferty@galton.uchicago.edu

## Abstract

We propose a simple, scalable, and fast gradient descent algorithm to optimize a nonconvex objective for the rank minimization problem and a closely related family of semidefinite programs. With $O(r^3 \kappa^2 n \log n)$ random measurements of a positive semidefinite $n \times n$ matrix of rank $r$ and condition number $\kappa$, our method is guaranteed to converge linearly to the global optimum.

## 1 Introduction

Semidefinite programming has become a key optimization tool in many areas of applied mathematics, signal processing and machine learning. SDPs often arise naturally from the problem structure, or are derived as surrogate optimizations that are relaxations of difficult combinatorial problems [7, 1, 8]. In spite of the importance of SDPs in principle—promising efficient algorithms with polynomial runtime guarantees—it is widely recognized that current optimization algorithms based on interior point methods can handle only relatively small problems. Thus, a considerable gap exists between the theory and applicability of SDP formulations. Scalable algorithms for semidefinite programming, and closely related families of nonconvex programs more generally, are greatly needed.

A parallel development is the surprising effectiveness of simple classical procedures such as gradient descent for large scale problems, as explored in the recent machine learning literature. In many areas of machine learning and signal processing such as classification, deep learning, and phase retrieval, gradient descent methods, in particular first order stochastic optimization, have led to remarkably efficient algorithms that can attack very large scale problems [3, 2, 10, 6]. In this paper we build on this work to develop first-order algorithms for solving the rank minimization problem under random measurements and a closely related family of semidefinite programs. Our algorithms are efficient and scalable, and we prove that they attain linear convergence to the global optimum under natural assumptions.

The affine rank minimization problem is to find a matrix $X^\star \in \mathbb{R}^{n \times p}$ of minimum rank satisfying constraints $\mathcal{A}(X^\star) = b$, where $\mathcal{A} : \mathbb{R}^{n \times p} \longrightarrow \mathbb{R}^m$ is an affine transformation. The underdetermined case where $m \ll np$ is of particular interest, and can be formulated as the optimization

$$
\begin{aligned}
\min_{X \in \mathbb{R}^{n \times p}} \quad & \mathrm{rank}(X) \\
\text{subject to} \quad & \mathcal{A}(X) = b.
\end{aligned}
\tag{1}
$$

This problem is a direct generalization of compressed sensing, and subsumes many machine learning problems such as image compression, low rank matrix completion and low-dimensional metric embedding [18, 12]. While the problem is natural and has many applications, the optimization is nonconvex and challenging to solve. Without conditions on the transformation $\mathcal{A}$ or the minimum rank solution $X^\star$, it is generally NP hard [15].

Existing methods, such as nuclear norm relaxation [18], singular value projection (SVP) [11], and alternating least squares (AltMinSense) [12], assume that a certain restricted isometry property (RIP) holds for $\mathcal{A}$. In the random measurement setting, this essentially means that at least $O(r(n + p)\log(n+p))$ measurements are available, where $r = \text{rank}(X^\star)$ [18]. In this work, we assume that (i) $X^\star$ is positive semidefinite and (ii) $\mathcal{A} : \mathbb{R}^{n \times n} \longrightarrow \mathbb{R}^m$ is defined as $\mathcal{A}(X)_i = \text{tr}(A_i X)$, where each $A_i$ is a random $n \times n$ symmetric matrix from the Gaussian Orthogonal Ensemble (GOE), with $(A_i)_{jj} \sim \mathcal{N}(0, 2)$ and $(A_i)_{jk} \sim \mathcal{N}(0, 1)$ for $j \neq k$. Our goal is thus to solve the optimization

$$\begin{aligned} \min_{X \succeq 0} \quad & \text{rank}(X) \\ \text{subject to} \quad & \text{tr}(A_i X) = b_i, \quad i = 1, \dots, m. \end{aligned} \tag{2}$$

In addition to the wide applicability of affine rank minimization, the problem is also closely connected to a class of semidefinite programs. In Section 2, we show that the minimizer of a particular class of SDP can be obtained by a linear transformation of $X^\star$. Thus, efficient algorithms for problem (2) can be applied in this setting as well.

Noting that a rank-$r$ solution $X^\star$ to (2) can be decomposed as $X^\star = Z^\star {Z^\star}^\top$ where $Z^\star \in \mathbb{R}^{n \times r}$, our approach is based on minimizing the squared residual

$$f(Z) = \frac{1}{4m} \left\| \mathcal{A}(ZZ^\top) - b \right\|^2 = \frac{1}{4m} \sum_{i=1}^{m} \left( \text{tr}(Z^\top A_i Z) - b_i \right)^2.$$

While this is a nonconvex function, we take motivation from recent work for phase retrieval by Candès et al. [6], and develop a gradient descent algorithm for optimizing $f(Z)$, using a carefully constructed initialization and step size. Our main contributions concerning this algorithm are as follows.

- We prove that with $O(r^3 n \log n)$ constraints our gradient descent scheme can exactly recover $X^\star$ with high probability. Empirical experiments show that this bound may potentially be improved to $O(rn \log n)$.
- We show that our method converges linearly, and has lower computational cost compared with previous methods.
- We carry out a detailed comparison of rank minimization algorithms, and demonstrate that when the measurement matrices $A_i$ are sparse, our gradient method significantly outperforms alternative approaches.

In Section 3 we briefly review related work. In Section 4 we discuss the gradient scheme in detail. Our main analytical results are presented in Section 5, with detailed proofs contained in the supplementary material. Our experimental results are presented in Section 6, and we conclude with a brief discussion of future work in Section 7.

## 2 Semidefinite Programming and Rank Minimization

Before reviewing related work and presenting our algorithm, we pause to explain the connection between semidefinite programming and rank minimization. This connection enables our scalable gradient descent algorithm to be applied and analyzed for certain classes of SDPs.

Consider a standard form semidefinite program

$$\begin{aligned} \min_{\widetilde{X} \succeq 0} \quad & \text{tr}(\widetilde{C}\widetilde{X}) \\ \text{subject to} \quad & \text{tr}(\widetilde{A}_i \widetilde{X}) = b_i, \quad i = 1, \dots, m \end{aligned} \tag{3}$$

where $\widetilde{C}, \widetilde{A}_1, \dots, \widetilde{A}_m \in \mathbb{S}^n$. If $\widetilde{C}$ is positive definite, then we can write $\widetilde{C} = LL^\top$ where $L \in \mathbb{R}^{n \times n}$ is invertible. It follows that the minimum of problem (3) is the same as

$$\begin{aligned} \min_{X \succeq 0} \quad & \text{tr}(X) \\ \text{subject to} \quad & \text{tr}(A_i X) = b_i, \quad i = 1, \dots, m \end{aligned} \tag{4}$$

where $A_i = L^{-1}\widetilde{A}_i {L^{-1}}^\top$. In particular, minimizers $\widetilde{X}^*$ of (3) are obtained from minimizers $X^*$ of (4) via the transformation

$$\widetilde{X}^* = {L^{-1}}^\top X^* L^{-1}.$$

Since $X$ is positive semidefinite, $\mathrm{tr}(X)$ is equal to $\|X\|_*$. Hence, problem (4) is the nuclear norm relaxation of problem (2). Next, we characterize the specific cases where $X^* = X^\star$, so that the SDP and rank minimization solutions coincide. The following result is from Recht et al. [18].

**Theorem 1.** *Let $\mathcal{A} : \mathbb{R}^{n \times n} \longrightarrow \mathbb{R}^m$ be a linear map. For every integer $k$ with $1 \leq k \leq n$, define the $k$-restricted isometry constant to be the smallest value $\delta_k$ such that*

$$(1 - \delta_k)\,\|X\|_F \leq \|\mathcal{A}(X)\| \leq (1 + \delta_k)\,\|X\|_F$$

*holds for any matrix $X$ of rank at most $k$. Suppose that there exists a rank $r$ matrix $X^\star$ such that $\mathcal{A}(X^\star) = b$. If $\delta_{2r} < 1$, then $X^\star$ is the only matrix of rank at most $r$ satisfying $\mathcal{A}(X) = b$. Furthermore, if $\delta_{5r} < 1/10$, then $X^\star$ can be attained by minimizing $\|X\|_*$ over the affine subset.*

In other words, since $\delta_{2r} \leq \delta_{5r}$, if $\delta_{5r} < 1/10$ holds for the transformation $\mathcal{A}$ and one finds a matrix $X$ of rank $r$ satisfying the affine constraint, then $X$ must be positive semidefinite. Hence, one can ignore the semidefinite constraint $X \succeq 0$ when solving the rank minimization (2). The resulting problem then can be exactly solved by nuclear norm relaxation. Since the minimum rank solution is positive semidefinite, it then coincides with the solution of the SDP (4), which is a constrained nuclear norm optimization.

The observation that one can ignore the semidefinite constraint justifies our experimental comparison with methods such as nuclear norm relaxation, `SVP`, and `AltMinSense`, described in the following section.

## 3  Related Work

Burer and Monteiro [4] proposed a general approach for solving semidefinite programs using factored, nonconvex optimization, giving mostly experimental support for the convergence of the algorithms. The first nontrivial guarantee for solving affine rank minimization problem is given by Recht et al. [18], based on replacing the rank function by the convex surrogate nuclear norm, as already mentioned in the previous section. While this is a convex problem, solving it in practice is nontrivial, and a variety of methods have been developed for efficient nuclear norm minimization. The most popular algorithms are proximal methods that perform singular value thresholding [5] at every iteration. While effective for small problem instances, the computational expense of the SVD prevents the method from being useful for large scale problems.

Recently, Jain et al. [11] proposed a projected gradient descent algorithm `SVP` (Singular Value Projection) that solves

$$\min_{X \in \mathbb{R}^{n \times p}} \quad \|\mathcal{A}(X) - b\|^2$$

$$\text{subject to} \quad \mathrm{rank}(X) \leq r,$$

where $\|\cdot\|$ is the $\ell_2$ vector norm and $r$ is the input rank. In the $(t+1)$th iteration, `SVP` updates $X^{t+1}$ as the best rank $r$ approximation to the gradient update $X^t - \mu \mathcal{A}^\top (\mathcal{A}(X^t) - b)$, which is constructed from the SVD. If $\mathrm{rank}(X^\star) = r$, then `SVP` can recover $X^\star$ under a similar RIP condition as the nuclear norm heuristic, and enjoys a linear numerical rate of convergence. Yet `SVP` suffers from the expensive per-iteration SVD for large problem instances.

Subsequent work of Jain et al. [12] proposes an alternating least squares algorithm `AltMinSense` that avoids the per-iteration SVD. `AltMinSense` factorizes $X$ into two factors $U \in \mathbb{R}^{n \times r}, V \in \mathbb{R}^{p \times r}$ such that $X = UV^\top$ and minimizes the squared residual $\|\mathcal{A}(UV^\top) - b\|^2$ by updating $U$ and $V$ alternately. Each update is a least squares problem. The authors show that the iterates obtained by `AltMinSense` converge to $X^\star$ linearly under a RIP condition. However, the least squares problems are often ill-conditioned, it is difficult to observe `AltMinSense` converging to $X^\star$ in practice.

As described above, considerable progress has been made on algorithms for rank minimization and certain semidefinite programming problems. Yet truly efficient, scalable and provably convergent

algorithms have not yet been obtained. In the specific setting that $X^\star$ is positive semidefinite, our algorithm exploits this structure to achieve these goals. We note that recent and independent work of Tu et al. [21] proposes a hybrid algorithm called *Procrustes Flow* (PF), which uses a few iterations of SVP as initialization, and then applies gradient descent.

## 4 A Gradient Descent Algorithm for Rank Minimization

Our method is described in Algorithm 1. It is parallel to the *Wirtinger Flow* (WF) algorithm for phase retrieval [6], to recover a complex vector $x \in \mathbb{C}^n$ given the squared magnitudes of its linear measurements $b_i = |\langle a_i, x \rangle|^2$, $i \in [m]$, where $a_1, \ldots, a_m \in \mathbb{C}^n$. Candès et al. [6] propose a first-order method to minimize the sum of squared residuals

$$f_{\text{WF}}(z) = \sum_{i=1}^{n} \left( |\langle a_i, z \rangle|^2 - b_i \right)^2. \tag{5}$$

The authors establish the convergence of WF to the global optimum—given sufficient measurements, the iterates of WF converge linearly to $x$ up to a global phase, with high probability.

If $z$ and the $a_i$s are real-valued, the function $f_{\text{WF}}(z)$ can be expressed as

$$f_{\text{WF}}(z) = \sum_{i=1}^{n} \left( z^\top a_i a_i^\top z - x^\top a_i a_i^\top x \right)^2,$$

which is a special case of $f(Z)$ where $A_i = a_i a_i^\top$ and each of $Z$ and $X^\star$ are rank one. See Figure 1a for an illustration; Figure 1b shows the convergence rate of our method. Our methods and results are thus generalizations of Wirtinger flow for phase retrieval.

Before turning to the presentation of our technical results in the following section, we present some intuition and remarks about how and why this algorithm works. For simplicity, let us assume that the rank is specified correctly.

Initialization is of course crucial in nonconvex optimization, as many local minima may be present. To obtain a sufficiently accurate initialization, we use a spectral method, similar to those used in [17, 6]. The starting point is the observation that a linear combination of the constraint values and matrices yields an unbiased estimate of the solution.

**Lemma 1.** *Let $M = \frac{1}{m} \sum_{i=1}^{m} b_i A_i$. Then $\frac{1}{2}\mathbb{E}(M) = X^\star$, where the expectation is with respect to the randomness in the measurement matrices $A_i$.*

Based on this fact, let $X^\star = U^\star \Sigma U^{\star \top}$ be the eigenvalue decomposition of $X^\star$, where $U^\star = [u_1^\star, \ldots, u_r^\star]$ and $\Sigma = \text{diag}(\sigma_1, \ldots, \sigma_r)$ such that $\sigma_1 \geq \ldots \geq \sigma_r$ are the nonzero eigenvalues of $X^\star$. Let $Z^\star = U^\star \Sigma^{\frac{1}{2}}$. Clearly, $u_s^\star = z_s^\star / \|z_s^\star\|$ is the top $s$th eigenvector of $\mathbb{E}(M)$ associated with eigenvalue $2 \|z_s^\star\|^2$. Therefore, we initialize according to $z_s^0 = \sqrt{\frac{|\lambda_s|}{2}} v_s$ where $(v_s, \lambda_s)$ is the top $s$th eigenpair of $M$. For sufficiently large $m$, it is reasonable to expect that $Z^0$ is close to $Z^\star$; this is confirmed by concentration of measure arguments.

Certain key properties of $f(Z)$ will be seen to yield a linear rate of convergence. In the analysis of convex functions, Nesterov [16] shows that for unconstrained optimization, the gradient descent scheme with sufficiently small step size will converge linearly to the optimum if the objective function is strongly convex and has a Lipschitz continuous gradient. However, these two properties are global and do not hold for our objective function $f(Z)$. Nevertheless, we expect that similar conditions hold for the local area near $Z^\star$. If so, then if we start close enough to $Z^\star$, we can achieve the global optimum.

In our subsequent analysis, we establish the convergence of Algorithm 1 with a constant step size of the form $\mu / \|Z^\star\|_F^2$, where $\mu$ is a small constant. Since $\|Z^\star\|_F$ is unknown, we replace it by $\|Z^0\|_F$.

## 5 Convergence Analysis

In this section we present our main result analyzing the gradient descent algorithm, and give a sketch of the proof. To begin, note that the symmetric decomposition of $X^\star$ is not unique, since

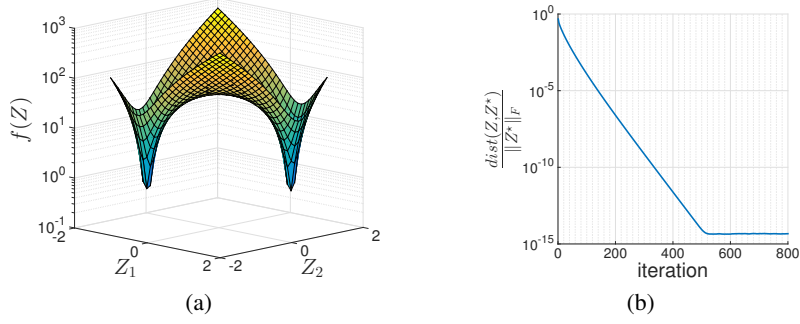

(a)　　　　　　　　　　　　(b)

Figure 1: (a) An instance of $f(Z)$ where $X^\star \in \mathbb{R}^{2\times 2}$ is rank-1 and $Z \in \mathbb{R}^2$. The underlying truth is $Z^\star = [1,1]^\top$. Both $Z^\star$ and $-Z^\star$ are minimizers. (b) Linear convergence of the gradient scheme, for $n = 200$, $m = 1000$ and $r = 2$. The distance metric is given in Definition 1.

---

**Algorithm 1:** Gradient descent for rank minimization

---

**input**: $\{A_i, b_i\}_{i=1}^m, r, \mu$

**initialization**

　　Set $(v_1, \lambda_1), \ldots, (v_r, \lambda_r)$ to the top $r$ eigenpairs of $\frac{1}{m}\sum_{i=1}^m b_i A_i$ s.t. $|\lambda_1| \geq \cdots \geq |\lambda_r|$

　　$Z^0 = [z_1^0, \ldots, z_r^0]$ where $z_s^0 = \sqrt{\frac{|\lambda_s|}{2}} \cdot v_s,\ s \in [r]$

　　$k \leftarrow 0$

**repeat**

$$\nabla f(Z^k) = \frac{1}{m}\sum_{i=1}^m \left(\mathrm{tr}(Z^{k\top} A_i Z^k) - b_i\right) A_i Z^k$$

$$Z^{k+1} = Z^k - \frac{\mu}{\sum_{s=1}^r |\lambda_s|/2}\nabla f(Z^k)$$

　　$k \leftarrow k + 1$

**until** *convergence*;

**output**: $\widehat{X} = Z^k Z^{k\top}$

---

$X^\star = (Z^\star U)(Z^\star U)^\top$ for any $r \times r$ orthonormal matrix $U$. Thus, the solution set is

$$\mathcal{S} = \left\{\widetilde{Z} \in \mathbb{R}^{n\times r} \mid \widetilde{Z} = Z^\star U \ \text{ for some } U \text{ with } UU^\top = U^\top U = I\right\}.$$

Note that $\|\widetilde{Z}\|_F^2 = \|X^\star\|_*$ for any $\widetilde{Z} \in \mathcal{S}$. We define the distance to the optimal solution in terms of this set.

**Definition 1.** *Define the distance between $Z$ and $Z^\star$ as*

$$d(Z, Z^\star) = \min_{UU^\top = U^\top U = I} \|Z - Z^\star U\|_F = \min_{\widetilde{Z}\in\mathcal{S}} \|Z - \widetilde{Z}\|_F.$$

Our main result for exact recovery is stated below, assuming that the rank is correctly specified. Since the true rank is typically unknown in practice, one can start from a very low rank and gradually increase it.

**Theorem 2.** *Let the condition number $\kappa = \sigma_1/\sigma_r$ denote the ratio of the largest to the smallest nonzero eigenvalues of $X^\star$. There exists a universal constant $c_0$ such that if $m \geq c_0\kappa^2 r^3 n \log n$, with high probability the initialization $Z^0$ satisfies*

$$d(Z^0, Z^\star) \leq \sqrt{\tfrac{3}{16}\sigma_r}. \tag{6}$$

*Moreover, there exists a universal constant $c_1$ such that when using constant step size $\mu/\|Z^\star\|_F^2$ with $\mu \leq \dfrac{c_1}{\kappa n}$ and initial value $Z^0$ obeying (6), the $k$th step of Algorithm 1 satisfies*

$$d(Z^k, Z^\star) \leq \sqrt{\tfrac{3}{16}\sigma_r}\left(1 - \frac{\mu}{12\kappa r}\right)^{k/2}$$

*with high probability.*

We now outline the proof, giving full details in the supplementary material. The proof has four main steps. The first step is to give a regularity condition under which the algorithm converges linearly if we start close enough to $Z^\star$. This provides a local regularity property that is similar to the Nesterov [16] criteria that the objective function is strongly convex and has a Lipschitz continuous gradient.

**Definition 2.** *Let $\overline{Z} = \arg\min_{\widetilde{Z}\in\mathcal{S}} \left\| Z - \widetilde{Z} \right\|_F$ denote the matrix closest to $Z$ in the solution set. We say that $f$ satisfies the regularity condition $RC(\varepsilon, \alpha, \beta)$ if there exist constants $\alpha$, $\beta$ such that for any $Z$ satisfying $d(Z, Z^\star) \leq \varepsilon$, we have*

$$\langle \nabla f(Z), Z - \overline{Z} \rangle \geq \frac{1}{\alpha}\sigma_r \left\| Z - \overline{Z} \right\|_F^2 + \frac{1}{\beta \left\| Z^\star \right\|_F^2} \left\| \nabla f(Z) \right\|_F^2 .$$

Using this regularity condition, we show that the iterative step of the algorithm moves closer to the optimum, if the current iterate is sufficiently close.

**Theorem 3.** *Consider the update $Z^{k+1} = Z^k - \dfrac{\mu}{\left\| Z^\star \right\|_F^2} \nabla f(Z^k)$. If $f$ satisfies $RC(\varepsilon, \alpha, \beta)$, $d(Z^k, Z^\star) \leq \varepsilon$, and $0 < \mu < \min(\alpha/2, 2/\beta)$, then*

$$d(Z^{k+1}, Z^\star) \leq \sqrt{1 - \frac{2\mu}{\alpha\kappa r}} d(Z^k, Z^\star).$$

In the next step of the proof, we condition on two events that will be shown to hold with high probability using concentration results. Let $\delta$ denote a small value to be specified later.

**A1**      For any $u \in \mathbb{R}^n$ such that $\|u\| \leq \sqrt{\sigma_1}$,    $\left\| \frac{1}{m}\sum_{i=1}^{m}(u^\top A_i u)A_i - 2uu^\top \right\| \leq \frac{\delta}{r}$.

**A2**      For any $\widetilde{Z} \in \mathcal{S}$,    $\left\| \dfrac{\partial^2 f(\widetilde{Z})}{\partial \widetilde{z}_s \partial \widetilde{z}_k^\top} - \mathbb{E}\left[ \dfrac{\partial^2 f(\widetilde{Z})}{\partial \widetilde{z}_s \partial \widetilde{z}_k^\top} \right] \right\| \leq \frac{\delta}{r}$,    for all $s, k \in [r]$.

Here the expectations are with respect to the random measurement matrices. Under these assumptions, we can show that the objective satisfies the regularity condition with high probability.

**Theorem 4.** *Suppose that **A1** and **A2** hold. If $\delta \leq \frac{1}{16}\sigma_r$, then $f$ satisfies the regularity condition $RC(\sqrt{\frac{3}{16}\sigma_r}, 24, 513\kappa n)$ with probability at least $1 - mCe^{-\rho n}$, where $C$, $\rho$ are universal constants.*

Next we show that under **A1**, a good initialization can be found.

**Theorem 5.** *Suppose that **A1** holds. Let $\{v_s, \lambda_s\}_{s=1}^{r}$ be the top $r$ eigenpairs of $M = \frac{1}{m}\sum_{i=1}^{m} b_i A_i$ such that $|\lambda_1| \geq \cdots \geq |\lambda_r|$. Let $Z^0 = [z_1, \ldots, z_r]$ where $z_s = \sqrt{\frac{|\lambda_s|}{2}} \cdot v_s$, $s \in [r]$. If $\delta \leq \frac{\sigma_r}{4\sqrt{r}}$, then*

$$d(Z^0, Z^\star) \leq \sqrt{3\sigma_r/16}.$$

Finally, we show that conditioning on **A1** and **A2** is valid since these events have high probability as long as $m$ is sufficiently large.

**Theorem 6.** *If the number of samples $m \geq \dfrac{42}{\min(\delta^2/r^2\sigma_1^2,\ \delta/r\sigma_1)} n\log n$, then for any $u \in \mathbb{R}^n$ satisfying $\|u\| \leq \sqrt{\sigma_1}$,*

$$\left\| \frac{1}{m}\sum_{i=1}^{m}(u^\top A_i u)A_i - 2uu^\top \right\| \leq \frac{\delta}{r}$$

*holds with probability at least $1 - mCe^{-\rho n} - \frac{2}{n^2}$, where $C$ and $\rho$ are universal constants.*

**Theorem 7.** *For any $x \in \mathbb{R}^n$, if $m \geq \dfrac{128}{\min(\delta^2/4r^2\sigma_1^2,\ \delta/2r\sigma_1)} n\log n$, then for any $\widetilde{Z} \in \mathcal{S}$*

$$\left\| \frac{\partial^2 f(\widetilde{Z})}{\partial \widetilde{z}_s \partial \widetilde{z}_k^\top} - \mathbb{E}\left[ \frac{\partial^2 f(\widetilde{Z})}{\partial \widetilde{z}_s \partial \widetilde{z}_k^\top} \right] \right\| \leq \frac{\delta}{r}, \quad \text{for all } s, k \in [r],$$

*with probability at least $1 - 6me^{-n} - \frac{4}{n^2}$.*

Note that since we need $\delta \leq \min\left(\frac{1}{16}, \frac{1}{4\sqrt{r}}\right)\sigma_r$, we have $\frac{\delta}{r\sigma_1} \leq 1$, and the number of measurements required by our algorithm scales as $O(r^3\kappa^2 n \log n)$, while only $O(r^2\kappa^2 n \log n)$ samples are required by the regularity condition. We conjecture this bound could be further improved to be $O(rn \log n)$; this is supported by the experimental results presented below.

Recently, Tu et al. [21] establish a tighter $O(r^2\kappa^2 n)$ bound overall. Specifically, when only one SVP step is used in preprocessing, the initialization of PF is also the spectral decomposition of $\frac{1}{2}M$. The authors show that $O(r^2\kappa^2 n)$ measurements are sufficient for $Z^0$ to satisfy $d(Z^0, Z^\star) \leq O(\sqrt{\sigma_r})$ with high probability, and demonstrate an $O(rn)$ sample complexity for the regularity condition.

# 6 Experiments

In this section we report the results of experiments on synthetic datasets. We compare our gradient descent algorithm with nuclear norm relaxation, SVP and AltMinSense for which we drop the positive semidefiniteness constraint, as justified by the observation in Section 2. We use ADMM for the nuclear norm minimization, based on the algorithm for the mixture approach in Tomioka et al. [19]; see Appendix G. For simplicity, we assume that AltMinSense, SVP and the gradient scheme know the true rank. Krylov subspace techniques such as the Lanczos method could be used compute the partial eigendecomposition; we use the randomized algorithm of Halko et al. [9] to compute the low rank SVD. All methods are implemented in MATLAB and the experiments were run on a MacBook Pro with a 2.5GHz Intel Core i7 processor and 16 GB memory.

## 6.1 Computational Complexity

It is instructive to compare the per-iteration cost of the different approaches; see Table 1. Suppose that the density (fraction of nonzero entries) of each $A_i$ is $\rho$. For AltMinSense, the cost of solving the least squares problem is $O(mn^2r^2 + n^3r^3 + mn^2r\rho)$. The other three methods have $O(mn^2\rho)$ cost to compute the affine transformation. For the nuclear norm approach, the $O(n^3)$ cost is from the SVD and the $O(m^2)$ cost is due to the update of the dual variables. The gradient scheme requires $2n^2r$ operations to compute $Z^k Z^{k^\top}$ and to multiply $Z^k$ by $n \times n$ matrix to obtain the gradient. SVP needs $O(n^2r)$ operations to compute the top $r$ singular vectors. However, in practice this partial SVD is more expensive than the $2n^2r$ cost required for the matrix multiplies in the gradient scheme.

| Method | Complexity |
|---|---|
| nuclear norm minimization via ADMM | $O(mn^2\rho + m^2 + n^3)$ |
| gradient descent | $O(mn^2\rho) + 2n^2r$ |
| SVP | $O(mn^2\rho + n^2r)$ |
| AltMinSense | $O(mn^2r^2 + n^3r^3 + mn^2r\rho)$ |

Table 1: Per-iteration computational complexities of different methods.

Clearly, AltMinSense is the least efficient. For the other approaches, in the dense case ($\rho$ large), the affine transformation dominates the computation. Our method removes the overhead caused by the SVD. In the sparse case ($\rho$ small), the other parts dominate and our method enjoys a low cost.

## 6.2 Runtime Comparison

We conduct experiments for both dense and sparse measurement matrices. AltMinSense is indeed slow, so we do not include it here.

In the first scenario, we randomly generate a $400 \times 400$ rank-2 matrix $X^\star = xx^\top + yy^\top$ where $x, y \sim \mathcal{N}(0, I)$. We also generate $m = 6n$ matrices $A_1, \ldots, A_m$ from the GOE, and then take $b = \mathcal{A}(X^\star)$. We report the relative error measured in the Frobenius norm defined as $\|\widehat{X} - X^\star\|_F / \|X^\star\|_F$. For the nuclear norm approach, we set the regularization parameter to $\lambda = 10^{-5}$. We test three values $\eta = 10, 100, 200$ for the penalty parameter and select $\eta = 100$ as it leads to the fastest convergence. Similarly, for SVP we evaluate the three values $5 \times 10^{-5}, 10^{-4}, 2 \times 10^{-4}$ for the step size, and select $10^{-4}$ as the largest for which SVP converges. For our approach, we test the three values $0.6, 0.8, 1.0$ for $\mu$ and select $0.8$ in the same way.

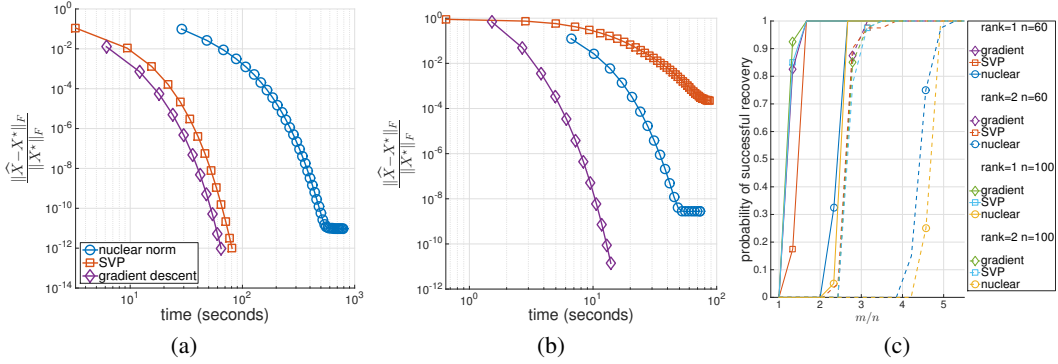

Figure 2: (a) Runtime comparison where $X^\star \in \mathbb{R}^{400 \times 400}$ is rank-2 and $A_i$s are dense. (b) Runtime comparison where $X^\star \in \mathbb{R}^{600 \times 600}$ is rank-2 and $A_i$s are sparse. (c) Sample complexity comparison.

In the second scenario, we use a more general and practical setting. We randomly generate a rank-2 matrix $X^\star \in \mathbb{R}^{600 \times 600}$ as before. We generate $m = 7n$ sparse $A_i$s whose entries are i.i.d. Bernoulli:

$$(A_i)_{jk} = \begin{cases} 1 & \text{with probability } \rho, \\ 0 & \text{with probability } 1 - \rho, \end{cases}$$

where we use $\rho = 0.001$. For all the methods we use the same strategies as before to select parameters. For the nuclear norm approach, we try three values $\eta = 10, 100, 200$ and select $\eta = 100$. For SVP, we test the three values $5 \times 10^{-3}, 2 \times 10^{-3}, 10^{-3}$ for the step size and select $10^{-3}$. For the gradient algorithm, we check the three values $0.8, 1, 1.5$ for $\mu$ and choose $1$.

The results are shown in Figures 2a and 2b. In the dense case, our method is faster than the nuclear norm approach and slightly outperforms SVP. In the sparse case, it is significantly faster than the other approaches.

## 6.3 Sample Complexity

We also evaluate the number of measurements required by each method to exactly recover $X^\star$, which we refer to as the *sample complexity*. We randomly generate the true matrix $X^\star \in \mathbb{R}^{n \times n}$ and compute the solutions of each method given $m$ measurements, where the $A_i$s are randomly drawn from the GOE. A solution with relative error below $10^{-5}$ is considered to be successful. We run 40 trials and compute the empirical probability of successful recovery.

We consider cases where $n = 60$ or $100$ and $X^\star$ is of rank one or two. The results are shown in Figure 2c. For SVP and our approach, the phase transitions happen around $m = 1.5n$ when $X^\star$ is rank-1 and $m = 2.5n$ when $X^\star$ is rank-2. This scaling is close to the number of degrees of freedom in each case; this confirms that the sample complexity scales linearly with the rank $r$. The phase transition for the nuclear norm approach occurs later. The results suggest that the sample complexity of our method should also scale as $O(rn \log n)$ as for SVP and the nuclear norm approach [11, 18].

## 7 Conclusion

We connect a special case of affine rank minimization to a class of semidefinite programs with random constraints. Building on a recently proposed first-order algorithm for phase retrieval [6], we develop a gradient descent procedure for rank minimization and establish convergence to the optimal solution with $O(r^3 n \log n)$ measurements. We conjecture that $O(rn \log n)$ measurements are sufficient for the method to converge, and that the conditions on the sampling matrices $A_i$ can be significantly weakened. More broadly, the technique used in this paper—factoring the semidefinite matrix variable, recasting the convex optimization as a nonconvex optimization, and applying first-order algorithms—first proposed by Burer and Monteiro [4], may be effective for a much wider class of SDPs, and deserves further study.

### Acknowledgements
Research supported in part by NSF grant IIS-1116730 and ONR grant N00014-12-1-0762.

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
