[Supplementary Material]

# A  Proof of Lemma 1

Let $A = (a_{ij})$ be a random matrix that is GOE distributed; thus $a_{ij} \sim \mathcal{N}(0,1)$ for $i \neq j$ and $a_{ii} \sim \mathcal{N}(0,2)$. We have $\mathbb{E}(M) = \sum_{s=1}^{r} \mathbb{E}((z_s^{\star\top} A z_s^{\star})A)$. Hence, it suffices to show that $\mathbb{E}((x^{\top}Ax)A) = 2xx^{\top}$ for any $x \in \mathbb{R}^n$. The $(i,j)$ entry of $(x^{\top}Ax)A$ has expected value

$$
\begin{aligned}
\mathbb{E}((x^{\top}Ax)a_{ij}) &= \mathbb{E}\left(\sum_k \sum_l x_k x_l a_{kl} a_{ij}\right) \\
&= \sum_k \sum_l x_k x_l \mathbb{E}(a_{kl} a_{ij}) \\
&= \sum_k \sum_l x_k x_l \cdot \begin{cases} 0 & \text{if } (k,l) \neq (i,j) \wedge (k,l) \neq (j,i) \\ \mathbb{E}(a_{kl}^2) & \text{otherwise} \end{cases} \\
&= \begin{cases} 2x_i x_j \mathbb{E}(a_{ij}^2) & \text{if } i \neq j \\ x_i^2 \mathbb{E}(a_{ii}^2) & \text{otherwise} \end{cases} \\
&= \begin{cases} 2x_i x_j & \text{if } i \neq j, \\ 2x_i^2 & \text{otherwise,} \end{cases}
\end{aligned}
$$

where we use that the variance of $a_{ii}$ is 2 and the variance of $a_{ij}$ is 1 for any $i \neq j$. In matrix form, this is $\mathbb{E}((x^{\top}Ax)A) = 2xx^{\top}$.

# B  Ingredients

We first present some technical lemmas that will be needed later. Recall Definition 2 that for any $Z$, $\overline{Z} = \arg\min_{\widetilde{Z} \in \mathcal{S}} \left\|Z - \widetilde{Z}\right\|_F$. Let $H = Z - \overline{Z}$. The $s$th column of $Z$, $\overline{Z}$, $Z^{\star}$, $H$ are denoted by $z_s$, $\bar{z}_s$, $z_s^{\star}$, $h_s$ respectively. We shall use the following formulas for the gradient and second order partial derivatives:

$$
\nabla f(Z) = \frac{1}{m} \sum_{i=1}^{m} \left(\text{tr}(H^{\top} A_i H) + 2\,\text{tr}(\overline{Z}^{\top} A_i H)\right)(A_i H + A_i \overline{Z}),
$$

$$
\frac{\partial^2 f(Z)}{\partial z_s \partial z_s^{\top}} = \frac{1}{m} \sum_{i=1}^{m} \left(2 A_i z_s z_s^{\top} A_i^{\top} + \left(\text{tr}(Z^{\top} A_i Z) - b_i\right) A_i\right), \quad \forall s \in [r],
$$

$$
\frac{\partial^2 f(Z)}{\partial z_s \partial z_k^{\top}} = \frac{1}{m} \sum_{i=1}^{m} 2 A_i z_s z_k^{\top} A_i^{\top}, \quad \forall s, k \in [r] \text{ such that } s \neq k.
$$

The next ingredient we need is the expectation of the second order partial derivatives with respect to the random measurement matrices.

**Lemma 2.** *Let $A = (a_{ij})$ be a GOE distributed random matrix. For any two fixed vectors $x$ and $y$, we have $\mathbb{E}\left[AxyA\right] = x^{\top}yI + yx^{\top}$.*

*Proof.* The expectation of $(i,j)$ entry of $Axy^{\top}A$ is

$$
\mathbb{E}[(Axy^{\top}A)_{ij}] = \mathbb{E}\left(\sum_{k\,l} a_{ik} a_{jk} x_k y_l\right).
$$

If $i = j$, then we have

$$
\mathbb{E}[(Axy^{\top}A)_{ii}] = \mathbb{E}\left(\sum_k a_{ik}^2 x_k y_k\right) = \sum_k x_k y_k + x_i y_i,
$$

since $\mathbb{V}\text{ar}(a_{ii}^2) = 2$ and $\mathbb{V}\text{ar}(a_{ik}^2) = 1$ if $k \neq i$. On the other hand, if $i \neq j$, then

$$
\mathbb{E}[(Axy^{\top}A)_{ij}] = \mathbb{E}\left(\sum_{kl} a_{ik} a_{jl} x_k y_l\right) = \mathbb{E}(a_{ij}^2 x_j y_i) = x_j y_i.
$$

Therefore, $\mathbb{E}(Axy^\top A) = x^\top y I + yx^\top$. $\qquad \square$

**Lemma 3.** *For all $s \in [r]$, it holds that $\mathbb{E}\left[\dfrac{\partial^2 f(Z)}{\partial z_s \partial z_s^\top}\right] = 2\left\|z_s\right\|^2 I + 2z_s z_s^\top + 2ZZ^\top - 2X^\star$ and $\mathbb{E}\left[\dfrac{\partial^2 f(Z)}{\partial z_s \partial z_k^\top}\right] = 2z_s^\top z_k I + 2z_k z_s^\top$ for all $k \in [r]$ such that $k \neq s$, where the expectation is over the random measurement matrices.*

*Proof.* The case where $k \neq s$ is a direct result of Lemma 2. For the other case, let $A = (a_{ij})$ be a GOE distributed random matrix. It follows from Lemma 1 that

$$\mathbb{E}\left[\dfrac{\partial^2 f(Z)}{\partial z_s \partial z_s^\top}\right] = 2\mathbb{E}(Az_s z_s^\top A) + 2ZZ^\top - 2X^\star.$$

By Lemma 2, we have

$$\mathbb{E}(Az_s z_s^\top A) = \left\|z_s\right\|^2 I + z_s z_s^\top.$$

Substituting this back into the above equation, we obtain the lemma. $\qquad \square$

We next recall a concentration result for the operator (spectral) norm of the random measurement matrices.

**Lemma 4.** *(Ledoux and Rider [14, Theorem 1]) There exists two absolute constants $C$ and $\rho = \frac{1}{\sqrt{8C}}$ such that with probability at least $1 - Ce^{-\rho n}$,*

$$\left\|A_i\right\| \leq 3\sqrt{n}.$$

A tighter upper bound is actually given in the *Tracy-Widow law*: w.h.p. $\left\|A_i\right\| = O(2\sqrt{n} + n^{1/6})$.

**Corollary 1.** *With probability at least $1 - mCe^{-\rho n}$, the average of the squared operator norm of the random measurement matrices is upper bounded by $9n$.*

*Proof.* Applying a union bound we have

$$\mathbb{P}\left(\frac{1}{m}\sum_{i=1}^m \left\|A_i\right\|^2 \leq 9n\right) \geq \mathbb{P}\left(\forall i, \left\|A_i\right\| \leq 3\sqrt{n}\right)$$

$$\geq 1 - \sum_{i=1}^m \mathbb{P}\left(\left\|A_i\right\| > 3\sqrt{n}\right)$$

$$\geq 1 - mCe^{-\rho n},$$

where we use Lemma 4 in the last line. $\qquad \square$

The following two technical lemmas are important tools for us. Define the set

$$E(\varepsilon) = \{Z \mid d(Z, Z^\star) \leq \varepsilon\}.$$

**Lemma 5.** *Suppose that A1 holds: $\left\|\frac{1}{m}\sum_{i=1}^m (u^\top A_i u)A_i - 2uu^\top\right\| \leq \frac{\delta}{r}$, for any $u$ such that $\|u\| \leq \sqrt{\sigma_1}$. If $\delta \leq \frac{1}{16}\sigma_r$, then for any $Z \in E\left(\sqrt{\frac{3}{16}\sigma_r}\right)$ it holds that*

$$2\left\|HH^\top\right\|_F^2 - \delta\|H\|_F^2 \leq \frac{1}{m}\sum_{i=1}^m tr(H^\top A_i H)^2 \leq \delta\|H\|_F^2 + 2\left\|HH^\top\right\|_F^2.$$

*Proof.* Let $h_s$ be the $s$th column of $H$. Since $\max_{s\in[r]}\|h_s\|_2 \leq \|H\|_F \leq \sqrt{\frac{3}{16}\sigma_r} \leq \sqrt{\sigma_1}$, it follows from the assumption of the lemma that

$$\left\|\frac{1}{m}\sum_{i=1}^m (h_s^\top A_i h_s)A_i - 2h_s h_s^\top\right\| \leq \frac{\delta}{r}, \quad s = 1, \ldots, r.$$

By the triangle inequality, we have

$$\left\| \frac{1}{m} \sum_{i=1}^{m} \sum_{s=1}^{r} (h_s^\top A_i h_s) A_i - 2 \sum_{s=1}^{r} h_s h_s^\top \right\| \le \delta$$

and consequently

$$-\delta \|h_s\|^2 \le h_s^\top \left( \frac{1}{m} \sum_{i=1}^{m} \operatorname{tr}(H^\top A_i H) A_i - 2HH^\top \right) h_s^\top \le \delta \|h_s\|^2, \ s = 1, \ldots, r,$$

where we replace $\sum_{s=1}^{r} h_s^\top A_i h_s$ by $\operatorname{tr}(H^\top A_i H)$ and $\sum_{s=1}^{r} h_s h_s^\top$ by $HH^\top$. Taking the sum of the above inequalities, we obtain

$$-\delta \|H\|_F^2 \le \frac{1}{m} \sum_{i=1}^{m} \operatorname{tr}(H^\top A_i H)^2 - 2 \operatorname{tr}(H^\top HH^\top H) \le \delta \|H\|_F^2.$$

Note that $\operatorname{tr}(H^\top HH^\top H) = \|HH^\top\|_F^2$. Therefore,

$$2 \|HH^\top\|_F^2 - \delta \|H\|_F^2 \le \frac{1}{m} \sum_{i=1}^{m} \operatorname{tr}(H^\top A_i H)^2 \le \delta \|H\|_F^2 + 2 \|HH^\top\|_F^2.$$

<div style="text-align:right">□</div>

**Lemma 6.** *Suppose that A2 holds: for any $\widetilde{Z}$ such that $\widetilde{Z}\widetilde{Z}^\top = X^\star$ we have*

$$\left\| \frac{\partial^2 f(\widetilde{Z})}{\partial \widetilde{z}_s \partial \widetilde{z}_k^\top} - \mathbb{E}\left[ \frac{\partial^2 f(\widetilde{Z})}{\partial \widetilde{z}_s \partial \widetilde{z}_k^\top} \right] \right\| \le \frac{\delta}{r}, \ \ s, k = 1, \ldots, r. \tag{7}$$

*Then*

$$\left( \sigma_r - \frac{\delta}{2} \right) \|H\|_F^2 + \|H^\top \bar{Z}\|_F^2 \le \frac{1}{m} \sum_{i=1}^{m} \operatorname{tr}(H^\top A_i \bar{Z})^2 \le \left( \sigma_1 + \frac{\delta}{2} \right) \|H\|_F^2 + \|H^\top \bar{Z}\|_F^2.$$

*Proof.* Our goal is to bound $\frac{1}{m} \sum_{i=1}^{m} \operatorname{tr}(H^\top A_i \bar{Z})^2$. This can be expanded as

$$\frac{1}{m} \sum_{i=1}^{m} \left( \sum_{s=1}^{r} (h_s^\top A_i \bar{z}_s) \right)^2 = \frac{1}{m} \sum_{i=1}^{m} \sum_{s=1}^{r} (h_s^\top A_i x_s)^2 + \frac{1}{m} \sum_{i=1}^{m} \sum_{s<k} 2(h_s^\top A_i x_s)(h_k^\top A_i x_k).$$

We first bound the sum of the quadratic terms. For any $s \in [r]$, we have

$$\frac{\partial^2 f(\bar{Z})}{\partial \bar{z}_s \partial \bar{z}_s^\top} = \frac{1}{m} \sum_{i=1}^{m} 2 A_i \bar{z}_s \bar{z}_s^\top A_i,$$

$$\mathbb{E}\left[ \frac{\partial^2 f(\bar{Z})}{\partial \bar{z}_s \partial \bar{z}_s^\top} \right] = 2 \|\bar{z}_s\|^2 I + 2\bar{z}_s \bar{z}_s^\top.$$

It follows from assumption (7) that for any $s \in [r]$,

$$-\frac{\delta}{r} \|h_s\|^2 \le \frac{1}{m} \sum_{i=1}^{m} 2(h_s^\top A_i \bar{z}_s)^2 - 2 \|\bar{z}_s\|^2 \|h_s\|^2 - 2(h_s^\top \bar{z}_s)^2 \le \frac{\delta}{r} \|h_s\|^2.$$

Taking the sum of above inequalities, we obtain

$$-\frac{\delta}{2r} \sum_{s=1}^{r} \|h_s\|^2 \le \frac{1}{m} \sum_{i=1}^{m} \sum_{s=1}^{r} (h_s^\top A_i \bar{z}_s)^2 - \sum_{s=1}^{r} \|\bar{z}_s\|^2 \|h_s\|^2 - \sum_{s=1}^{r} (h_s^\top \bar{z}_s)^2 \le \frac{\delta}{2r} \sum_{s=1}^{r} \|h_s\|^2. \tag{8}$$

Similarly, we bound the sum of the cross terms. For any fixed $s, k$ such that $s \neq k$, we have

$$\frac{\partial^2 f(\overline{Z})}{\partial \overline{z}_s \partial \overline{z}_k^\top} = \frac{1}{m} f(\overline{Z}) \sum_{i=1}^m 2 A_i \overline{z}_s \overline{z}_k^\top A_i,$$

$$\mathbb{E}\left[\frac{\partial^2 f(\overline{Z})}{\partial \overline{z}_s \partial \overline{z}_k^\top}\right] = 2 \overline{z}_s^\top \overline{z}_k I + 2 \overline{z}_k \overline{z}_s^\top,$$

and consequently

$$-\frac{\delta}{r} \sum_{s<k} \|h_s\| \|h_k\| \leq \frac{1}{m} \sum_{i=1}^m \sum_{s<k} 2(h_s^\top A_i \overline{z}_s)(h_k^\top A_i \overline{z}_k) - 2 \sum_{s<k} \overline{z}_s^\top \overline{z}_k h_s^\top h_k - 2 \sum_{s<k} h_s^\top \overline{z}_k \overline{z}_s^\top h_k \tag{9}$$

$$\leq \frac{\delta}{r} \sum_{s<k} \|h_s\| \|h_k\|.$$

We combine equations (9) and (8) to get

$$-\frac{\delta}{2r} \sum_{sk} \|h_s\| \|h_k\| \leq \frac{1}{m} \sum_{i=1}^m \operatorname{tr}(H^\top A_i \overline{Z})^2 - \sum_{sk} \overline{z}_s^\top \overline{z}_k h_s^\top h_k - \sum_{sk} h_s^\top \overline{z}_k \overline{z}_s^\top h_k \leq \frac{\delta}{2r} \sum_{sk} \|h_s\| \|h_k\|. \tag{10}$$

Note that $\sum_{sk} h_s^\top \overline{z}_k \overline{z}_s^\top h_k = \operatorname{tr}(H^\top \overline{Z} H^\top \overline{Z})$, $\sum_{sk} \overline{z}_s^\top \overline{z}_k h_s^\top h_k = \left\|\overline{Z} H^\top\right\|_F^2$ and

$$\sum_{sk} \|h_s\| \|h_k\| = \left(\sum_{s=1}^r \|h_s\|\right)^2 \leq r \sum_{s=1}^r \|h_s\|^2 = r \|H\|_F^2.$$

By Lemma 7, $\operatorname{tr}(H^\top \overline{Z} H^\top \overline{Z}) = \left\|H^\top \overline{Z}\right\|_F^2$. Replacing those terms in equation (10) gives us

$$-\frac{\delta}{2} \|H\|_F^2 + \left\|\overline{Z} H^\top\right\|_F^2 + \left\|H^\top \overline{Z}\right\|_F^2 \leq \frac{1}{m} \sum_{i=1}^m \operatorname{tr}(H^\top A_i \overline{Z})^2 \leq \frac{\delta}{2} \|H\|_F^2 + \left\|\overline{Z} H^\top\right\|_F^2 + \left\|H^\top \overline{Z}\right\|_F^2.$$

Finally, we obtain the claim by noticing that

$$\sqrt{\sigma_r} \|H\|_F \leq \left\|\overline{Z} H^\top\right\|_F \leq \sqrt{\sigma_1} \|H\|_F,$$

where $\sqrt{\sigma_1} = \sigma_{\max}(\overline{Z}) \geq \cdots \geq \sigma_{\min}(\overline{Z}) = \sqrt{\sigma_r}$ are the singular values of $\overline{Z}$. $\qquad\square$

**Lemma 7.** $tr(H^\top \overline{Z} H^\top \overline{Z}) = \left\|H^\top \overline{Z}\right\|_F^2.$

*Proof.* Let $\bar{U} = \arg\min_{UU^\top = U^\top U = I} \|Z - Z^\star U\|_F^2 = \arg\max_{UU^\top = U^\top U = I} \langle U, Z^{\star\top} Z \rangle$. Note that $\langle A, B \rangle \leq \|A\|_* \|B\|$ for any matrices $A, B$ that are of the same size. The equality holds when $B = U_A V_A^\top$ where $A = U_A \Sigma_A V_A^\top$ is the SVD of $A$. Hence, $\bar{U} = \widetilde{U}\widetilde{V}^\top$ where $\widetilde{U}\widetilde{S}\widetilde{V}^\top$ is the SVD of $Z^{\star\top} Z$; $\overline{Z} = Z^\star \bar{U}$. Therefore, $\overline{Z}^\top \overline{Z} = Z^\top Z^\star \bar{U} = \widetilde{V}\widetilde{S}\widetilde{V}^\top$ is symmetric and positive semidefinite. Thus, $H^\top \overline{Z} = \overline{Z}^\top \overline{Z} - \overline{Z}^\top \overline{Z}$ is also symmetric. This implies that $\operatorname{tr}(H^\top \overline{Z} H^\top \overline{Z}) = \left\|H^\top \overline{Z}\right\|_F^2$. $\qquad\square$

# C  Linear Convergence

**Proof of Theorem 3**

Let $H^k = Z^k - \overline{Z}^k$. Then we have that

$$
\begin{aligned}
\left\| Z^{k+1} - \overline{Z}^k \right\|_F^2 &= \left\| Z^k - \frac{\mu}{\|Z^\star\|_F^2} \nabla f(Z^k) - \overline{Z}^k \right\|_F^2 \\
&= \left\| H^k \right\|_F^2 + \frac{\mu^2}{\|Z^\star\|_F^4} \left\| \nabla f(Z^k) \right\|_F^2 - \frac{2\mu}{\|Z^\star\|_F^2} \langle \nabla f(Z^k), H^k \rangle \\
&\leq \left\| H^k \right\|_F^2 + \frac{\mu^2}{\|Z^\star\|_F^4} \left\| \nabla f(Z^k) \right\|_F^2 - \frac{2\mu}{\|Z^\star\|_F^2} \left( \frac{1}{\alpha} \sigma_r \left\| H^k \right\|_F^2 + \frac{1}{\beta \|Z^\star\|_F^2} \left\| \nabla f(Z^k) \right\|_F^2 \right) \\
&= \left( 1 - \frac{2\mu}{\alpha} \cdot \frac{\sigma_r}{\sum_{s=1}^r \sigma_s} \right) \left\| H^k \right\|_F^2 + \frac{\mu(\mu - 2/\beta)}{\|Z^\star\|_F^4} \left\| \nabla f(Z^k) \right\|_F^2 \\
&\leq \left( 1 - \frac{2\mu}{\alpha} \cdot \frac{\sigma_r}{r\sigma_1} \right) \left\| H^k \right\|_F^2 \\
&= \left( 1 - \frac{2\mu}{\alpha \kappa r} \right) d(Z^k, Z^\star)^2,
\end{aligned}
$$

where we use the definition of $RC(\varepsilon, \alpha, \beta)$ in the third line, $\|Z^\star\|_F^2 = \|X^\star\|_* = \sum_{s=1}^r \sigma_s$ in the third to last line and $0 < \mu < \min\{\alpha/2, 2/\beta\}$ in the second to last line. Therefore,

$$
d(Z^{k+1}, Z^\star) = \min_{\widetilde{Z} \in \mathcal{S}} \left\| Z^{k+1} - \widetilde{Z} \right\|_F^2 \leq \sqrt{1 - \frac{2\mu}{\alpha \kappa r}} d(Z^k, Z^\star).
$$

# D  Regularity Condition

As mentioned before, Nesterov [16, Theorem 2.1.11] shows that the gradient scheme converges linearly under a condition similar to the regularity condition, which is satisfied if the function is strongly convex and has a Lipschitz continuous gradient (*strongly smooth*). In order to prove Theorem 4, we show that with high probability the function $f$ satisfies the local curvature condition, which is analogous to strong convexity, and the local smoothness condition, which is analogous to strong smoothness.

### C1  *Local Curvature Condition*

There exists a constant $C_1$ such that for any $Z$ satisfying $d(Z, Z^\star) \leq \sqrt{\frac{3}{16} \sigma_r}$,

$$
\langle \nabla f(Z), Z - \overline{Z} \rangle \geq C_1 \left\| Z - \overline{Z} \right\|_F^2 + \left\| (Z - \overline{Z})^\top \overline{Z} \right\|_F^2.
$$

### C2  *Local Smoothness Condition*

There exist constants $C_2, C_3$ such that for any $Z$ satisfying $d(Z, Z^\star) \leq \sqrt{\frac{3}{16} \sigma_r}$,

$$
\|\nabla f(Z)\|_F^2 \leq C_2 \left\| Z - \overline{Z} \right\|_F^2 + C_3 \left\| (Z - \overline{Z})^\top \overline{Z} \right\|_F^2.
$$

## D.1 Proof of the Local Curvature Condition

$$
\begin{aligned}
\langle \nabla f(Z), H \rangle = \ & \overbrace{\frac{2}{m}\sum_{i=1}^{m}\operatorname{tr}(H^\top A_i \bar{Z})^2}^{p^2} + \overbrace{\frac{1}{m}\sum_{i=1}^{m}\operatorname{tr}(H^\top A_i H)^2}^{q^2} + \frac{3}{m}\sum_{i=1}^{m}\operatorname{tr}(H^\top A_i \bar{Z})\operatorname{tr}(H^\top A_i H) \\
\geq \ & p^2 + q^2 - \frac{3}{m}\sqrt{\sum_{i=1}^{m}\operatorname{tr}(H^\top A_i \bar{Z})^2}\sqrt{\sum_{i=1}^{m}\operatorname{tr}(H^\top A_i H)^2} \\
= \ & p^2 + q^2 - \frac{3}{\sqrt{2}}\overbrace{\sqrt{\frac{2}{m}\sum_{i=1}^{m}\operatorname{tr}(H^\top A_i \bar{Z})^2}}^{p}\overbrace{\sqrt{\frac{1}{m}\sum_{i=1}^{m}\operatorname{tr}(H^\top A_i H)^2}}^{q} \\
= \ & \left(p - \frac{3}{2\sqrt{2}}q\right)^2 - \frac{1}{8}q^2 \\
\geq \ & \left(\frac{p^2}{2} - \frac{9}{8}q^2\right) - \frac{1}{8}q^2 \\
= \ & \frac{p^2}{2} - \frac{5}{4}q^2 = \frac{1}{m}\sum_{i=1}^{m}\operatorname{tr}(H^\top A_i \bar{Z})^2 - \frac{5}{4}\frac{1}{m}\sum_{i}\operatorname{tr}(H^\top A_i H)^2 \\
\geq \ & \left(\sigma_r - \frac{\delta}{2}\right)\|H\|_F^2 + \left\|H^\top \bar{Z}\right\|_F^2 - \frac{5\delta}{4}\|H\|_F^2 - \frac{5}{2}\left\|HH^\top\right\|_F^2 \\
\geq \ & \left(\sigma_r - \frac{5}{2}\|H\|_F^2 - \frac{7}{4}\delta\right)\|H\|_F^2 + \left\|H^\top \bar{Z}\right\|_F^2 .
\end{aligned}
$$

where we use Cauchy-Schwarz inequality in the 2nd line, the inequality $(a - b)^2 \geq \frac{a^2}{2} - b^2$ in the 5th line, Lemma 5 and 6 in the 7th line, and the fact that $\left\|HH^\top\right\|_F \leq \|H\|_F^2$ in the 8th line. Since $\|H\|_F \leq \sqrt{\frac{3}{16}\sigma_r}$ and $\delta \leq \frac{1}{16}\sigma_r$, we have

$$
\langle \nabla f(Z), H \rangle \geq \frac{27}{64}\sigma_r \|H\|_F^2 + \left\|H^\top \bar{Z}\right\|_F^2 . \tag{11}
$$

## D.2 Proof of the Local Smoothness Condition

We need to upper bound $\|\nabla f(Z)\|_F^2 = \max_{\|W\|_F = 1}|\langle \nabla f(Z), W\rangle|^2$. It suffices to show that for any $W \in \mathbb{R}^{n \times R}$ of unit Frobenius norm, $|\langle \nabla f(Z), W\rangle|^2$ is upper bounded if $Z \in E\left(\sqrt{\frac{3}{16}\sigma_r}\right)$. Since $(a + b + c + d)^2 \leq 4(a^2 + b^2 + c^2 + d^2)$, we have

$$
\begin{aligned}
|\langle \nabla f(Z), W\rangle|^2 = \ & \left(\frac{1}{m}\sum_{i=1}^{m}\left(\operatorname{tr}(H^\top A_i H) + 2\operatorname{tr}(H^\top A_i \bar{Z})\right)\left(\operatorname{tr}(W^\top A_i H) + \operatorname{tr}(W^\top A_i \bar{Z})\right)\right)^2 \\
= \ & \left(\frac{1}{m}\sum_{i=1}^{m}\operatorname{tr}(H^\top A_i H)\operatorname{tr}(W^\top A_i H) + 2\operatorname{tr}(H^\top A_i \bar{Z})\operatorname{tr}(W^\top A_i H)\right. \\
& \left. + \operatorname{tr}(H^\top A_i H)\operatorname{tr}(W^\top A_i \bar{Z}) + 2\operatorname{tr}(H^\top A_i \bar{Z})\operatorname{tr}(W^\top A_i \bar{Z})\right)^2 \\
\leq \ & 4\left(\frac{1}{m}\sum_{i=1}^{m}\operatorname{tr}(H^\top A_i H)\operatorname{tr}(W^\top A_i H)\right)^2 + 4\left(\frac{2}{m}\sum_{i=1}^{m}\operatorname{tr}(H^\top A_i \bar{Z})\operatorname{tr}(W^\top A_i H)\right)^2 \\
& + 4\left(\frac{1}{m}\sum_{i=1}^{m}\operatorname{tr}(H^\top A_i H)\operatorname{tr}(W^\top A_i \bar{Z})\right)^2 + 4\left(\frac{2}{m}\sum_{i=1}^{m}\operatorname{tr}(H^\top A_i \bar{Z})\operatorname{tr}(W^\top A_i \bar{Z})\right)^2 .
\end{aligned}
$$

The first term in the righthand side can be upper bounded as

$$4\left(\frac{1}{m}\sum_{i=1}^{m}\operatorname{tr}(H^{\top}A_iH)\operatorname{tr}(W^{\top}A_iH)\right)^2 \le 4\left(\frac{1}{m}\sum_{i=1}^{m}\operatorname{tr}(H^{\top}A_iH)^2\right)\left(\frac{1}{m}\sum_{i=1}^{m}\operatorname{tr}(W^{\top}A_iH)^2\right)$$

$$\le 4\left(2\|H\|_F^4+\delta\|H\|_F^2\right)\left(\frac{1}{m}\sum_{i=1}^{m}\|W\|_F^2\|A_iH\|_F^2\right)$$

$$= 4\left(2\|H\|_F^4+\delta\|H\|_F^2\right)\left(\frac{1}{m}\sum_{i=1}^{m}\|A_iH\|_F^2\right)$$

$$\le 4\left(2\|H\|_F^4+\delta\|H\|_F^2\right)\left(\frac{1}{m}\sum_{i=1}^{m}\|A_i\|^2\|H\|_F^2\right)$$

$$\le 36n\|H\|_F^2\left(2\|H\|_F^4+\delta\|H\|_F^2\right),$$

where we use the Cauchy-Schwarz inequality in the first and second line, Lemma 5 and $\left\|HH^{\top}\right\|_F \le \|H\|_F^2$ in the third line and Corollary 1 in the last line.

The other three terms are bounded similarly. For the second term, we have

$$4\left(\frac{2}{m}\sum_{i=1}^{m}\operatorname{tr}(H^{\top}A_i\overline{Z})\operatorname{tr}(W^{\top}A_iH)\right)^2 \le 16\left(\frac{1}{m}\sum_{i=1}^{m}\operatorname{tr}(H^{\top}A_i\overline{Z})^2\right)\left(\frac{1}{m}\sum_{i=1}^{m}\operatorname{tr}(W^{\top}A_iH)^2\right)$$

$$\le 36n\|H\|_F^2\left((4\sigma_1+2\delta)\|H\|_F^2+4\left\|H^{\top}\overline{Z}\right\|_F^2\right),$$

where we use Lemma 6 and 1. The third term is bounded as

$$4\left(\frac{1}{m}\sum_{i=1}^{m}\operatorname{tr}(H^{\top}A_iH)\operatorname{tr}(W^{\top}A_i\overline{Z})\right)^2 \le 4\left(\frac{1}{m}\sum_{i=1}^{m}\operatorname{tr}(H^{\top}A_iH)^2\right)\left(\frac{1}{m}\sum_{i=1}^{m}\operatorname{tr}(W^{\top}A_i\overline{Z})^2\right)$$

$$\le 36n\left\|\overline{Z}\right\|_F^2\left(2\|H\|_F^4+\delta\|H\|_F^2\right),$$

and the fourth term is bounded as

$$4\left(\frac{2}{m}\sum_{i=1}^{m}\operatorname{tr}(H^{\top}A_i\overline{Z})\operatorname{tr}(W^{\top}A_i\overline{Z})\right)^2 \le 16\left(\frac{1}{m}\sum_{i=1}^{m}\operatorname{tr}(H^{\top}A_i\overline{Z})^2\right)\left(\frac{1}{m}\sum_{i=1}^{m}(W^{\top}A_i\overline{Z})^2\right)$$

$$\le 36n\left\|\overline{Z}\right\|_F^2\left((4\sigma_1+2\delta)\|H\|_F^2+4\left\|H^{\top}\overline{Z}\right\|_F^2\right).$$

Putting these inequalities together, we have

$$\|\nabla f(Z)\|_F^2 \le 36n\left(\left\|\overline{Z}\right\|_F^2+\|H\|_F^2\right)\left(2\|H\|_F^4+(4\sigma_1+3\delta)\|H\|_F^2+4\left\|H^{\top}\overline{Z}\right\|_F^2\right).$$

Hence,

$$\frac{\|\nabla f(Z)\|_F^2}{144n\left(\left\|\overline{Z}\right\|_F^2+\|H\|_F^2\right)} \le \left(\sigma_1+\frac{1}{2}\|H\|_F^2+\frac{3}{4}\delta\right)\|H\|_F^2+\left\|H^{\top}\overline{Z}\right\|_F^2.$$

Since $\|H\|_F \le \sqrt{\frac{3}{16}\sigma_r}$ and $\delta \le \frac{1}{16}\sigma_r$, we have

$$\frac{\|\nabla f(Z)\|^2}{144n\left(\left\|\overline{Z}\right\|_F^2+(3/16)\sigma_r\right)} \le \left(\sigma_1+\frac{9}{64}\sigma_r\right)\|H\|_F^2+\left\|H^{\top}\overline{Z}\right\|_F^2.$$

### D.3   Proof of the Regularity Condition

Now we combine the curvature and the smoothness conditions. For any $\gamma \in \left(0,\frac{\sigma_1}{\sigma_r}\right)$, it holds that

$$\gamma\frac{\sigma_r}{\sigma_1}\cdot\frac{\|\nabla f(Z)\|_F^2}{144n\left(\left\|\overline{Z}\right\|_F^2+(3/16)\sigma_r\right)} \le \gamma\frac{\sigma_r}{\sigma_1}\cdot\left(\sigma_1+\frac{9}{64}\sigma_r\right)\|H\|_F^2+\left\|H^{\top}\overline{Z}\right\|_F^2. \qquad (12)$$

Combining equation ([11](#)) and ([12](#)), we obtain

$$\langle \nabla f(Z), H\rangle \geq \left(\frac{27}{64} - \gamma - \gamma\frac{\sigma_r}{\sigma_1}\frac{9}{64}\right)\sigma_r \|H\|_F^2 + \gamma\frac{\sigma_r}{\sigma_1} \cdot \frac{\|\nabla f(Z)\|_F^2}{144n(\|\bar{Z}\|_F^2 + (3/16)\sigma_r)}$$

$$\geq \left(\frac{27}{64} - \frac{73}{64}\gamma\right)\sigma_r \|H\|_F^2 + \gamma\frac{\sigma_r}{\sigma_1} \cdot \frac{\|\nabla f(Z)\|_F^2}{144n(\|\bar{Z}\|_F^2 + (3/16)\sigma_r)}.$$

If we take $\gamma = \frac{1}{3}$, then

$$\langle \nabla f(Z), H\rangle \geq \frac{1}{24}\sigma_r \|H\|_F^2 + \frac{\sigma_r}{\sigma_1} \cdot \frac{\|\nabla f(Z)\|_F^2}{3 \cdot 144n\left(\|\bar{Z}\|_F^2 + (3/16)\sigma_r\right)}$$

$$\geq \frac{1}{24}\sigma_r \|H\|_F^2 + \frac{\sigma_r/\sigma_1}{513n\|Z^\star\|_F^2}\|\nabla f(Z)\|_F^2,$$

where we use $\|\bar{Z}\|_F^2 = \|Z^\star\|_F^2 = \|X^\star\|_* \geq \sigma_r$. Thus we have

$$\langle \nabla f(Z), H\rangle \geq \frac{1}{\alpha}\sigma_r \|H\|_F^2 + \frac{1}{\beta\|Z^\star\|_F^2}\|\nabla f(Z)\|_F^2$$

for $\alpha \geq 24$ and $\beta \geq \frac{\sigma_1}{\sigma_r} \cdot 513n$.

# E    Initialization

**Proof of Theorem [5](#)**

By assumption, we have

$$\left\|\frac{1}{m}\sum_{i=1}^m (z_s^{\star\top} A_i z_s^\star)A_i - 2z_s^\star z_s^{\star\top}\right\| \leq \frac{\delta}{r}, \quad s \in [r].$$

Hence,

$$\|M - 2X^\star\| = \left\|\frac{1}{m}\sum_{i=1}^m\sum_{s=1}^r (z_s^{\star\top} A_i z_s^\star)A_i - 2\sum_{s=1}^r z_s^\star z_s^{\star T}\right\| \leq \sum_{s=1}^r \left\|\frac{1}{m}\sum_{i=1}^m (z_s^{\star\top} A_i z_s^\star)A_i - 2z_s^\star z_s^{\star\top}\right\| \leq \delta. \tag{13}$$

Let $\lambda_1' \geq \cdots \geq \lambda_n'$ be the eigenvalues of $M$. By Weyl's theorem, we have

$$|\lambda_s' - 2\sigma_s| \leq \delta, \quad s \in [n].$$

Since $\delta < \sigma_r$, it is easy to see $\lambda_1' \geq \cdots \geq \lambda_r' > \delta$ and $|\lambda_s'| \leq \delta, s = r+1, \ldots, n$. Hence, $\lambda_s = \lambda_s'$, $s \in [r]$, and $Z^0 Z^{0\top}$ is the best rank $r$ approximation of $\frac{1}{2}M$. Therefore,

$$\left\|Z^0 Z^{0\top} - Z^\star Z^{\star\top}\right\|_F = \left\|Z^0 Z^{0\top} - \frac{1}{2}M + \frac{1}{2}M - Z^\star Z^{\star\top}\right\|_F$$

$$\leq \left\|Z^0 Z^{0\top} - \frac{1}{2}M\right\|_F + \left\|\frac{1}{2}M - Z^\star Z^{\star\top}\right\|_F$$

$$\leq 2\left\|\frac{1}{2}M - Z^\star Z^{\star\top}\right\|_F$$

$$\leq \sqrt{2r}\left\|M - 2Z^\star Z^{\star\top}\right\|$$

$$\leq \sqrt{2r}\delta.$$

where we used the fact $Z^0 Z^{0\top} = \arg\min_{\text{rank}(X)\leq r}\left\|X - Z^\star Z^{\star\top}\right\|_F$ in the third line, $\|A\|_F \leq \sqrt{\text{rank}(A)}\|A\|$ in the second to last line, and inequality ([13](#)) in the last line.

Let $H = Z^0 - \overline{Z}^0$. We want to bound $d(Z^0, Z^\star)^2 = \|H\|_F^2$. According to the discussion in Lemma 7, $H^\top \overline{Z}^0$ is symmetric and $Z^{0^\top}\overline{Z}^0$ is positive semidefinite.

The following step closely follows [21]. It holds that

$$
\begin{aligned}
\left\|Z^0 Z^{0^\top} - Z^\star Z^{\star^\top}\right\|_F^2 &= \left\|Z^0 Z^{0^\top} - \overline{Z}^0 \overline{Z}^{0^\top}\right\|_F^2 \\
&= \left\|H\overline{Z}^{0^\top} + \overline{Z}^0 H^\top + HH^\top\right\|_F^2 \\
&= \mathrm{tr}\Big(\overline{Z}^0 H^\top H \overline{Z}^{0^\top} + H\overline{Z}^{0^\top} H \overline{Z}^{0^\top} + HH^\top \overline{Z}^{0^\top} \\
&\qquad + \overline{Z}^0 H^\top \overline{Z}^0 H^\top + H\overline{Z}^{0^\top}\overline{Z}^0 H + HH^\top \overline{Z}^0 H^\top \\
&\qquad + \overline{Z}^0 H^\top HH^\top + H\overline{Z}^{0^\top} HH^\top + HH^\top HH^\top\Big) \\
&= \mathrm{tr}\Big((H^\top H)^2 + 2(H^\top \overline{Z}^0)^2 + 2(H^\top H)(\overline{Z}^{0^\top}\overline{Z}^0) + 4(H^\top H)(H^\top \overline{Z}^0)\Big) \\
&= \mathrm{tr}\Big(\Big(H^\top H + \sqrt{2}H^\top \overline{Z}^0\Big)^2 + (4 - 2\sqrt{2})(H^\top H)(H^\top \overline{Z}^0) + 2(H^\top H)(\overline{Z}^{0^\top}\overline{Z}^0)\Big) \\
&\geq \mathrm{tr}\Big((4 - 2\sqrt{2})(H^\top H)(H^\top \overline{Z}^0) + 2(H^\top H)(\overline{Z}^\top \overline{Z})\Big) \\
&= \mathrm{tr}\Big((4 - 2\sqrt{2})(H^\top H)(Z^{0^\top}\overline{Z}^0)\Big) + \mathrm{tr}\Big((2\sqrt{2} - 2)(H^\top H)(\overline{Z}^\top \overline{Z})\Big),
\end{aligned}
$$

where in the fourth line we used the property that the trace is invariant under cyclic permutations and $H^\top \overline{Z}^0 = \overline{Z}^{0^\top} H$.

Since $Z^{0^\top}\overline{Z}^0$ is positive semidefinite, $\mathrm{tr}((H^\top H)(Z^{0^\top}\overline{Z}^0))$ is nonnegative. Hence,

$$
\begin{aligned}
\left\|Z^0 Z^{0^\top} - Z^\star Z^{\star^\top}\right\|_F^2 &\geq (2\sqrt{2} - 2)\,\mathrm{tr}\left((H^\top H)(\overline{Z}^\top \overline{Z})\right) \\
&= (2\sqrt{2} - 2)\left\|H\overline{Z}^\top\right\|_F^2 \\
&\geq (2\sqrt{2} - 2)\|H\|_F^2\,\sigma_r \\
&= (2\sqrt{2} - 2)\sigma_r d(Z^0, Z^\star)^2.
\end{aligned}
$$

If $\delta \leq \frac{\sigma_r}{4\sqrt{r}}$, then

$$
d(Z^0, Z^\star)^2 \leq \frac{\left\|Z^0 Z^0 - Z^\star Z^{\star^\top}\right\|_F^2}{(2\sqrt{2} - 2)\sigma_r} \leq \frac{2r\delta^2}{(2\sqrt{2} - 2)\sigma_r} \leq \frac{3}{16}\sigma_r.
$$

# F  Sample Complexity

In this section, we verify that our assumptions hold with high probability if $m \geq cn\log n$, where $c$ is a constant that depends on $\delta$, $r$, and $\kappa$. Our proof relies on the following concentration inequality.

**Theorem 8.** *(Matrix Bernstein Inequality [20]) Let $S_1, \ldots, S_m$ be independent random matrices with dimension $n \times n$. Assume that $\mathbb{E}(S_i) = 0$ and $\|S_i\| \leq L$, for all $i \in [m]$. Let $\nu^2 = \max\left\{\left\|\sum_{i=1}^m \mathbb{E}(S_i S_i^\top)\right\|, \left\|\sum_{i=1}^m \mathbb{E}(S_i^\top S_i)\right\|\right\}$. Then for all $\delta \geq 0$,*

$$
\mathbb{P}\left(\left\|\frac{1}{m}\sum_{i=1}^m S_i\right\| \geq \delta\right) \leq 2n\exp\left(\frac{-m^2\delta^2}{\nu^2 + Lm\delta/3}\right).
$$

We first give a technical lemma that we will use later.

**Lemma 8.** *Let $A = (a_{ij})$ be a random matrix drawn from GOE. Let $S = a_{11}A - 2e_1 e_1^\top$. There exist absolute constants $C$, $\rho$ such that with probability at least $1 - Ce^{-\rho n}$, we have*

$$
\|S\| \leq 18n.
$$

*Proof.* Let $\widetilde{A} = A - a_{11}e_1e_1^\top$. $S = a_{11}\widetilde{A} + (a_{11}^2 - 2)e_1e_1^\top$. Note that $a_{11}$ and $\widetilde{A}$ are independent, hence $\|S\| \leq |a_{11}|\|\widetilde{A}\| + |a_{11}^2 - 2|$. Besides, since $a_{11} \sim \mathcal{N}(0, 2)$, we can see that $a_{11}^2/2$ is $\chi^2$ distributed.

First we bound the operator norm of $\widetilde{A}$. We rewrite $\|\widetilde{A}\|$ as

$$\|\widetilde{A}\| = \max_{\|u\|=1} |u^\top \widetilde{A}u| = \max_{\|u\|=1} |u^\top Du - du_1^2| \leq \|D\| + |d|,$$

where $D = \widetilde{A} + de_1e_1^\top$, $d \sim \mathcal{N}(0, 2)$. As $D$ is GOE distributed, by Lemma 4,

$$\mathbb{P}\left(\|D\| > 3\sqrt{n}\right) \leq C'e^{-\rho'n}, \tag{14}$$

where $C'$ and $\rho'$ are absolute constants.

Using the Gaussian tail inequality, we have

$$\mathbb{P}\left(|d| > 2\sqrt{n}\right) \leq 2e^{-n}. \tag{15}$$

Combining inequalities (14) and (15), we have

$$\mathbb{P}\left(\|\widetilde{A}\| > 5\sqrt{n}\right) \leq \mathbb{P}\left(\|D\| > 3\sqrt{n} \vee |d| > 2\sqrt{n}\right) \leq C'e^{-\rho'n} + 2e^{-n}, \tag{16}$$

where the last inequality follows from the union bound.

Next we bound the deviation of the $\chi^2$ term. By the corollary of Lemma 1 in Laurent and Massart [13], we have

$$\mathbb{P}(|a_{11}^2 - 2| > 4(\sqrt{n} + n)) \leq 2e^{-n}. \tag{17}$$

Since $a_{11}$ is identically distributed as $d$, inequality (15) holds for $a_{11}$ as well. Namely, $\mathbb{P}\left(|a_{11}| > 2\sqrt{n}\right) \leq 2e^{-n}$. Combining this with inequalities (17), (16), we have

$$\mathbb{P}\left(\|S\| \leq 14n + 4\sqrt{n}\right) \geq 1 - 6e^{-n} - C'e^{-\rho'n}.$$

Finally, the statement is obtained by choosing proper $C$, $\rho$, and using $\sqrt{n} \leq n$. $\qquad\square$

## F.1 Proof of Theorem 6

*Proof.* It is equivalent to show that for any unit vector $u$, with high probability,

$$\left\|\frac{1}{m}\sum_{i=1}^m (u^\top A_i u)A_i - 2uu^\top\right\| \leq \frac{\delta}{r\sigma_1}.$$

If $P$ is an orthonormal matrix, then

$$\left\|\frac{1}{m}\sum_{i=1}^m \left((Pu)^\top A_i(Pu)\right)A_i - 2(Pu)(Pu)^\top\right\| = \left\|\frac{1}{m}\sum_{i=1}^m \left(u^\top(P^\top A_i P)uA_i\right) - 2Puu^\top P^\top\right\|$$

$$= \left\|\frac{1}{m}\sum_{i=1}^m u^\top(P^\top A_i P)uP^\top A_i P - 2uu^\top\right\|$$

$$= \left\|\frac{1}{m}\sum_{i=1}^m u^\top \widetilde{A}_i u\widetilde{A}_i - 2uu^\top\right\|,$$

where in the second line we use unitary invariance of the operator norm, and in the last line we denote $P^\top A_i P$ by $\widetilde{A}_i$. Since the GOE is invariant under orthogonal conjugation, $\widetilde{A}_i$ and $A_i$ are identically distributed. Hence, it suffices to prove the claim when $u = e_1$, i.e.

$$\left\|\frac{1}{m}\sum_{i=1}^m a_{11}^{(i)} A_i - 2e_1e_1^\top\right\| \leq \delta_0,$$

where $a_{11}^{(i)}$ is the $(1, 1)$ entry of $A_i$ and $\delta_0 = \frac{\delta}{r\sigma_1}$.

To show this, we apply Theorem 8, where $S_i = a_{11}^{(i)} A_i - 2e_1 e_1^\top$. This requires that the operator norm of $S_i$ is bounded, for each $i$. We address this by noticing that with high probability $\|S_i\| \leq 18n$, $\forall i$. To be precise, by Lemma 8 there exist constants $C, \rho$, such that

$$\mathbb{P}\left(\|S_i\| > 18n\right) \leq Ce^{-\rho n}, \quad i = 1, \ldots, m.$$

Taking the union bound over all the $S_i$s leads to

$$\mathbb{P}\left(\max_i \|S_i\| > 18n\right) \leq mCe^{-\rho n}. \tag{18}$$

Next, we calculate $\nu^2 = \left\|\sum_{i=1}^m \mathbb{E}(S_i^2)\right\| = m \left\|\mathbb{E}(S_1^2)\right\|$. Let $A = (a_{ij})$ denote $A_1$, $S$ denote $S_1$. We have $\mathbb{E}(S^2) = \mathbb{E}(a_{11}{}^2 A^2) - 4e_1 e_1^\top$, and

$$\left(a_{11}^2 A^2\right)_{11} = a_{11}^4 + \sum_{k=2}^n a_{11}^2 a_{1k}^2,$$

$$\left(a_{11}^2 A^2\right)_{ii} = a_{11}^2 \left(a_{ii}^2 + \sum_{k \neq i}^n a_{ik}^2\right), \quad \forall i \neq 1,$$

$$\left(a_{11}^2 A^2\right)_{ij} = a_{11}^2 \sum_{k=1}^n a_{ik} a_{jk}, \quad \forall i \neq j.$$

It is easy to see that $\mathbb{E}(a_{11}^2 A^2) = \mathrm{diag}(2n+10, 2n+2, \ldots, 2n+2)$. Consequently, $\nu^2 = (2n+6)m$. By Theorem 8, if $m \geq \frac{42}{\min(\delta_0^2, \delta_0)} \cdot n \log n$, then

$$\begin{aligned}
\mathbb{P}\left(\left\|\frac{1}{m}\sum_{i=1}^m S_i\right\| \geq \delta_0\right) &\leq 2n \exp\left(\frac{-m\delta_0^2}{2n(1+3\delta_0)+6}\right) \\
&\leq 2n \exp\left(\frac{-m\delta_0^2}{2n(4+3\delta_0)}\right) \\
&\leq 2n \exp\left(\frac{-m\delta_0^2}{14n \cdot \max(1, \delta_0)}\right) \\
&\leq \frac{2}{n^2}.
\end{aligned} \tag{19}$$

Combining inequalities (18) and (19), we conclude that

$$\mathbb{P}\left(\left\|\frac{1}{m}\sum_{i=1}^m a_{11}^{(i)} A_i - 2e_1 e_1^\top\right\| \leq \delta_0\right) \geq 1 - mCe^{-\rho n} - \frac{2}{n^2}.$$

$\square$

### F.2 Proof of Theorem 7

The formulation of the second order partial derivatives and their expectations is given in Appendix B.

It is easy to see that for any $\overline{Z} \in \mathcal{S}$, $\max_{s \in [r]} \|\bar{z}_r\| \leq \sqrt{\sigma_1}$. Thus it is sufficient to prove that for any two unitary vector $u$ and $y$ with high probability it holds that

$$\left\|\frac{1}{m}\sum_{i=1}^m 2A_i uy^\top A_i - 2u^\top y I - 2yu^\top\right\| \leq \frac{\delta}{r\sigma_1}.$$

We can decompose $y$ as $y = \beta u + \beta_\perp u_\perp$ for a certain unit vector $u_\perp$ that is orthogonal to $u$, where $\beta^2 + \beta_\perp^2 = 1$. Let $\delta_0 = \frac{\delta}{2r\sigma_1}$. It suffices to prove the following two claims.

(i) For any unitary vector $u$, with high probability
$$\left\| \frac{1}{m} \sum_{i=1}^{m} 2A_i uu^\top A_i - 2I - 2uu^\top \right\| \leq \delta_0.$$

(ii) For any two orthogonal unit vectors $u$ and $u_\perp$, with high probability
$$\left\| \frac{1}{m} \sum_{i=1}^{m} 2A_i uu_\perp^\top A_i - 2u_\perp u^\top \right\| \leq \delta_0.$$

**Proof of (i)**

If $P$ is an orthonormal matrix, then
$$\left\| \frac{1}{m} \sum_{i=1}^{m} 2A_i Puu^\top PA_i - 2I - 2Puu^\top P^\top \right\| = \left\| \frac{1}{m} \sum_{i=1}^{m} 2P^\top A_i Puu^\top P^\top A_i P - 2I - 2uu^\top \right\|$$
$$= \left\| \frac{1}{m} \sum_{i=1}^{m} 2\widetilde{A}_i uu^\top \widetilde{A}_i - 2I - 2uu^\top \right\|,$$

where $\widetilde{A}_i$ and $A_i$ have the same distribution. Hence we only need to prove the case where $u = e_1$:
$$\left\| \frac{1}{m} \sum_{i=1}^{m} 2v^{(i)} v^{(i)\top} - 2I - 2e_1 e_1^\top \right\| \leq \delta_0,$$

where $v^{(i)} = A_i e_1$ is the first column of $A_i$.

Let $S_i = 2(v^{(i)} v^{(i)\top} - I - e_1 e_1^\top)$. To apply Theorem 8, we need to show that with high probability $\|S_i\|$ is bounded for each $i$ and calculate $\nu^2 = \left\| \sum_{i=1}^{n} \mathbb{E}(S_i^2) \right\| = m \left\| \mathbb{E}(S_1^2) \right\|$.

Let $S, v, A$ denote $S_1, v^{(1)}$, and $A^{(1)}$ respectively. It is easy to see that
$$\|S\| \leq 2\|v\|^2 + 4 = 2(w + a_{11}^2) + 4,$$
where $w = \sum_{k=2}^{n} a_{1k}^2$. As $a_{11} \sim \mathcal{N}(0, 2)$, $a_{1k} \sim \mathcal{N}(0, 1)$ for $k \neq 1$, we can see that $a_{11}^2/2$ and $w$ are $\chi^2$ distributed with degrees of freedom 1 and $n - 1$, respectively. Using the $\chi^2$ tail bound, we have
$$\mathbb{P}\left( a_{11}^2/2 > 2(\sqrt{n} + n) + 1 \right) \leq e^{-n},$$
$$\mathbb{P}\left( w > 5n - 1 \right) \leq e^{-n}, \quad k = 2, \ldots, n.$$

It follows from the union bound that
$$\mathbb{P}\left( \|S\| > 26n + 6 \right) \leq 2e^{-n},$$

and consequently
$$\mathbb{P}\left( \max_i \|S_i\| > 26n + 6 \right) \leq 2me^{-n}. \tag{20}$$

To calculate $\nu^2$, we expand $\mathbb{E}(S^2)$ as
$$\mathbb{E}(S^2) = 4\mathbb{E}\left( (vv^\top)^2 \right) - 4(I + e_1 e_1^\top)^2$$
$$= 4\mathbb{E}\left( \|v\|^2 vv^\top \right) - 4(I + 3e_1 e_1^\top).$$

Some simple calculations show that
$$\left( \|v\|^2 vv^\top \right)_{11} = v_1^4 + \sum_{k=2}^{n} v_k^2 v_1^2,$$
$$\left( \|v\|^2 vv^\top \right)_{jj} = v_1^2 v_j^2 + v_j^4 + \sum_{k \neq 1, j} v_k^2 v_j^2, \quad j = 2, \ldots, n,$$
$$\left( \|v\|^2 vv^\top \right)_{jl} = \sum_{k=1}^{n} v_k^2 v_j v_l, \quad j < l.$$

As $v_1 \sim \mathcal{N}(0,2)$, $v_j \sim \mathcal{N}(0,1)$ for $j \neq 1$,

$$\mathbb{E}\left(\|v\|^2 vv^\top\right)_{11} = 2n + 10,$$

$$\mathbb{E}\left(\|v\|^2 vv^\top\right)_{jj} = n + 3, \quad j = 2, \ldots, n,$$

$$\mathbb{E}\left(\|v\|^2 vv^\top\right)_{jl} = 0, \quad j < l.$$

Hence, $\mathbb{E}(S^2) = \mathrm{diag}(8n + 24, 4n + 8, \ldots, 4n + 8)$ and thus $\nu^2 = m(8n + 24)$.

If $m \geq (128/\min(\delta_0^2, \delta_0))n \log n$, then by applying Theorem 8 we can see

$$
\mathbb{P}\left(\left\|\frac{1}{m}\sum_{i=1}^m 2v^{(i)}v^{(i)\top} - 2I - 2e_1e_1^\top\right\| > \delta_0\right) \leq 2n \exp\left(\frac{-m\delta_0^2}{8n + 24 + (\frac{26}{3}n + 2)\delta_0}\right)
$$

$$
\leq 2n \exp\left(\frac{-m\delta_0^2}{(128/3)n\max(1,\delta_0)}\right) \qquad (21)
$$

$$
\leq \frac{2}{n^2}.
$$

Combining inequalities (21) and (20) leads to

$$
\mathbb{P}\left(\left\|\frac{1}{m}\sum_{i=1}^m 2v^{(i)}v^{(i)\top} - 2I - 2e_1e_1^\top\right\| \leq \delta_0\right) \geq 1 - 2me^{-n} - \frac{2}{n^2}.
$$

**Proof of (ii)**

We only need to prove the case where $u = e_1$ and $u_\perp = e_2$ due to the same reason above. That is,

$$
\left\|\frac{1}{m}\sum_{i=1}^m 2v^{(i)}q^{(i)\top} - 2e_2e_1^\top\right\| \leq \delta_0,
$$

where $v^{(i)}$ and $q^{(i)}$ are the first and second columns of $A_i$.

As before, let $S_i = 2(v^{(i)}q^{(i)\top} - e_2e_1^\top)$ and let $S, v, q, A$ denote $S_1, v^{(1)}, q^{(1)}$ and $A^{(1)}$ respectively. From the proof of (i), we can see that with probability at least $1 - 4e^{-n}$ both $\|v\|$ and $\|q\|$ are no larger than $\sqrt{13n + 1}$. Since $\|S\| \leq 2\|v\|\|q\| + 2$, we have

$$
\mathbb{P}\left(\max_i \|S_i\| \geq 26n + 4\right) \leq 4me^{-n}.
$$

Next, we calculate $\nu^2 = m \max\left\{\left\|\mathbb{E}(SS^\top)\right\|, \left\|\mathbb{E}(S^\top S)\right\|\right\}$.

$$\mathbb{E}(SS^\top) = 4\mathbb{E}(\|q\|^2)\mathbb{E}(vv^\top) + 4e_2e_2^\top.$$

$$\mathbb{E}(S^\top S) = 4\mathbb{E}(\|v\|^2)\mathbb{E}(qq^\top) + 4e_1e_1^\top.$$

Some simple calculation shows that $\mathbb{E}(\|v\|^2) = \mathbb{E}(\|q\|^2) = n + 1$, $\mathbb{E}(vv^\top) = I + e_1e_1^\top$ and $\mathbb{E}(qq^\top) = I + e_2e_2^\top$. Hence,

$$\mathbb{E}(SS^\top) = 4(n + 1)I + 4(n + 1)e_1e_1^\top + 4e_2e_2^\top,$$

$$\mathbb{E}(S^\top S) = 4(n + 1)I + 4(n + 1)e_2e_2^\top + 4e_1e_1^\top,$$

and $\nu^2 = 8(n + 1)m$. If $m \geq \frac{78}{\min(\delta_0^2, \delta_0)}n \log n$, then by applying Theorem 8 we have

$$
\mathbb{P}\left(\left\|\frac{1}{m}\sum_{i=1}^m 2v^{(i)}q^{(i)\top} - 2e_1e_2^\top\right\| > \delta_0\right) \leq 2n \exp\left(\frac{-m\delta_0^2}{8n + 8 + (\frac{26n+4}{3})\delta_0}\right)
$$

$$
\leq 2n \exp\left(\frac{-m\delta_0^2}{26n\max(1,\delta_0)}\right) \qquad (22)
$$

$$
\leq \frac{2}{n^2}.
$$

This means,

$$\mathbb{P}\left(\left\|\frac{1}{m}\sum_{i=1}^{m} 2v^{(i)}q^{(i)\top} - 2e_1 e_2^\top\right\| \le \delta_0\right) \ge 1 - 4me^{-n} - \frac{2}{n^2}.$$

## G  ADMM for Nuclear Norm Minimization

We reformulate the nuclear norm minimizing problem as

$$\min_{X \in \mathbb{R}^{n \times n}} \quad \frac{1}{2\lambda}\|\mathcal{A}(X) - b\|^2 + \|X\|_*,\tag{23}$$

where $\lambda > 0$ is the regularization parameter. $\lambda \to 0$ will enforce the minimizer $X_{\mathrm{nuc}}^*$ satisfying the affine constraint $\mathcal{A}(X_{\mathrm{nuc}}^*) = b$.

We apply ADMM to the dual problem of (23):

$$\begin{aligned}
\min_{\alpha \in \mathbb{R}^m, V \in \mathbb{R}^{n \times n}} \quad & \frac{\lambda}{2}\|\alpha\|^2 - \alpha^\top b \\
\text{subject to} \quad & \|V\| \le 1 \\
& \mathcal{A}^\top(\alpha) = V,
\end{aligned}\tag{24}$$

where we introduce an auxiliary variable $V$ to make this problem equality constrained.

The augmented Lagrangian of problem (24) can be written as

$$L_\eta(\alpha, X) = \frac{\lambda}{2}\|\alpha\|^2 - \alpha^\top b + \mathbf{1}_{\|\cdot\|\le 1}(V) + \langle X, \mathcal{A}^\top(\alpha) - V\rangle + \frac{\eta}{2}\left\|\mathcal{A}^\top(\alpha) - V\right\|_F^2,$$

where $X$ is the multiplier, $\eta$ is the penalty parameter, and $\mathbf{1}_{\|\cdot\|\le 1}$ is the indicator function of the unit spectral norm ball i.e. $\mathbf{1}_{\|\cdot\|\le 1}(V)$ equals 0 if $\|V\| \le 1$ and $+\infty$ otherwise.

Let $\mathrm{vec}(\cdot)$ denote the vectorization of a matrix, whose inverse mapping is denoted by $\mathrm{mat}(\cdot)$. We can rewrite the transformations as $\mathcal{A}(X) = \boldsymbol{A}\mathrm{vec}(X)$ and $\mathcal{A}^\top(\alpha) = \mathrm{mat}(\boldsymbol{A}^\top \alpha) = \sum_{i=1}^m \alpha_i A_i$, where $\boldsymbol{A}$ is a $m \times n^2$ matrix whose $i$th row is $\mathrm{vec}(A_i)^\top$.

The ADMM starts from initialization $(\alpha^0, V^0, X^0)$ and updates the three variables alternately. The updates can be computed in close forms:

$$\alpha^{k+1} = (\lambda I + \eta \boldsymbol{A}\boldsymbol{A}^\top)^{-1}\left(b + \boldsymbol{A}\mathrm{vec}\big(\eta V^k - X^k\big)\right),$$

$$V^{k+1} = \mathrm{proj}\left(\sum_{i=1}^m \alpha_i^{k+1} A_i + X^k/\eta\right),$$

$$X^{k+1} = X^k + \eta\left(\sum_{i=1}^m \alpha_i^{k+1} A_i - V^{k+1}\right),$$

where $\mathrm{proj}(\cdot)$ is the projection onto the unit spectral norm ball. Let $X = U\Sigma V^\top$ be the singular value decomposition of $X$,

$$\mathrm{proj}(X) = U\min(\Sigma, 1)V^\top.$$

In fact, the update of $V$ can be combined with other steps without being computed explicitly. One only has to iterate the following two steps:

$$\alpha^{k+1} = (\lambda I + \eta \boldsymbol{A}\boldsymbol{A}^\top)^{-1}\left(b + \boldsymbol{A}\mathrm{vec}\big(\eta\sum_{i=1}^m \alpha_i^k A_i + X^{k-1} - 2X^k\big)\right),$$

$$X^{k+1} = \mathrm{prox}_\eta\left(\eta\sum_{i=1}^m \alpha_i^{k+1} A_i + X^k\right),$$

where $\mathrm{prox}_\eta(\cdot)$ is the singular value soft-thresholding operator defined as

$$\mathrm{prox}_\eta(X) = U\max(\Sigma - \eta, 0)V^\top.$$

The sequence of multipliers $\{X^k\}$ converges to the primal solution of (23). To speed up the update of $\alpha$, the Cholesky decomposition of $\lambda I + \eta \boldsymbol{A}\boldsymbol{A}^\top$ is precomputed in our implementation.