[Reviews · NeurIPS 2015]

Submitted by Assigned_Reviewer_1

The paper studies gradient descent algorithms for rank minimisation subject to affine constraints. The main claim to fame is an analysis of number the number of samples required for exact recovery in a of the Gaussian ensemble setting (which resembles http://arxiv.org/abs/1212.0467).

I am in two minds, when I read the paper. One one hand, the paper is well written -- perhaps the best-written out of those I have reviewed this year -- and I trust all the claims. On the other hand, neither the theoretical nor the implementational aspects are particularly strong.

The theoretical results are based on the work of Jain (going all the way to http://arxiv.org/abs/1212.0467) and Candes (http://arxiv.org/abs/1407.1065) and are of incremental nature. The paper does attribute the credit, at least to Candes.

The implementation seems awful. When compared to recent implementations, e.g. http://arxiv.org/abs/1408.2467 the performance seems orders of magnitude away from the state of the art -- and being an order of magnitude faster than general-purpose SDP solver on the nuclear norm does not make it any better. The authors should acknowledge that and compare the results with other codes on some established benchmark (e.g. Lenna), so as to show that the price in terms of run-time brings about much better performance in terms of objective function values (SNR, RMSE) -- which is plausible, but far from certain.

Minor comments:

Introduction is misleading. The technical results come with many ifs and buts and the introduction is not making that clear enough.

Section 6.1 is mis-leading, because the sole fact that one method is more expensive per-iteration does not make it "least efficient", unless there is a very tight analysis of the rates of convergence parametrised by the same m, n, p, r (which there isn't), which the section does not make clear enough.
Summary: The paper is well written -- perhaps the best-written out of those I have reviewed this year -- and I trust all the claims, but neither the theoretical nor the implementational aspects are strong enough for a "strong accept".

Submitted by Assigned_Reviewer_2

I was asked to provide a "light review" of this work.

The paper presents results on recovery of low-rank semidefinite matrices from linear measurements, using nonconvex optimization. The approach is inspired by recent work on phase retrieval, and combines spectral initialization with gradient descent. The connection to phase retrieval comes because measurements which are linear in the semidefinite matrix X = Z Z' are quadratic in the factors Z. The paper proves recovery results which imply that correct recovery occurs when the number of measurements m is essentially proportional to n r^2, where n is the dimensionality and r is the rank. The convergence analysis is based on a form of restricted strong convexity (restricted because there is an r(r-1)/2-dimensional set of equivalent solutions along which the objective is flat). This condition also implies linear convergence of the proposed algorithm.

The paper contains novel results on rank minimization, and a novel technical approach. Understanding the properties of rank minimization problems that enable nonconvex methods to succeed is certainly of interest. However, the results are not completely decisive. The main issue is that the paper does not obtain order-optimal sample complexity - the results are suboptimal by a factor of r. Hence, the theoretical results are not really on par with the current best known results for efficient algorithms for this problem. Also, the results are restricted to semidefinite matrices, and hence misses many compelling machine learning applications of rank minimization.

The results could be compared to those of De Sa, Olukotun and Re (Global Convergence of Stochastic Gradient Descent for Some Non-convex Matrix Problems). This submission has a better dependence on the rank parameter, but requires spectral initialization.

In the experiments on nuclear norm minimization, the positive semidefinite constraint should not be dropped.
Summary: The paper contains novel results on rank minimization, and a novel technical approach. Understanding the properties of rank minimization problems that enable nonconvex methods to succeed is certainly of interest. However, the results are not completely decisive. The main issue is that the paper does not obtain order-optimal sample complexity - the results are suboptimal by a factor of r. Hence, the theoretical results are not really on par with the current best known results for efficient algorithms for this problem. Also, the results are restricted to semidefinite matrices, and hence misses many compelling machine learning applications of rank minimization.

Submitted by Assigned_Reviewer_3

++++++++++ Summary:

Pros:

Interesting approach and techniques Well written paper with clear exposition Overall idea could spur further improvements

Cons: References/comparisons to some important existing approaches are missing.

++++++++++ Detailed:

The authors propose an efficient algorithm for the recovery of a low-rank matrix from random affine measurements. The authors maintain a factorized representation of the target matrix; pose the recovery problem as a non-convex optimization involving the factored representation; and solve the optimization using a technique that they call "Wirtinger Flow". Essentially, the approach consists of a careful initialization followed by a series of gradient descent iterations. The descent converges linearly and the overall algorithm exhibits a similar asymptotic runtime as singular value projection (SVP). Numerical experiments support the authors' claim that their approach is faster than existing methods.

The paper is well written and the exposition is clear. The paper adds to the growing body of work in the algorithmic learning literature that all revolve around the "spectral initialization + gradient descent" type of idea. While the theoretical improvement over existing methods (such as SVP) is only by a constant, the empirical gains are evocative and the techniques developed in the paper could lead to significantly faster algorithms in the future.

One concern regarding prior work: the authors present their algorithm as a considerable improvement over nuclear norm minimization or SVP primarily due to the benefit of not having to incur an SVD per iteration.

However, they do not compare theoretical/numerical performance with the low-rank SDP approaches of Burer and Monteiro [05] and numerous follow-up works. Like the authors, these approaches also do not require an expensive SVD in each iteration; perform gradient descent; and converge rapidly to the optimum. How does the authors' technique improve over these methods?

Some other comments: - Consider also comparing with the Admira approach of Lee-Bresler '09. - The variable p is used to denote matrix dimension in (1), and sparsity of the measurement matrices in Table 1.

- Why state the results of Table 1 in terms of p? For the GOE (which is what the authors consider in their theoretical exposition), p = n^2 almost surely.
Summary: The paper proposes a scalable algorithm for recovering a low-rank matrix from affine measurements. The algorithm is similar to gradient descent with a careful spectral initialization, and exhibits provable linear convergence. While certain comparisons to the literature are missing, the contributions of the paper are compelling and the techniques developed here could spur several follow-up ideas.

Submitted by Assigned_Reviewer_4

This paper proposes a gradient descent type algorithm for rank minimization in the space of symmetric positive semidefinite matrices with linear equality constraints. It was motivated by the recent work by Candes et al [5]. The basic algorithm as in [5] is based on the Wirtinger flow algorithm. Convergence analysis and sample complexity analysis are also provided for the proposed algorithm. The paper is well-written with consistent notations and a high level outline of the convergence analysis. These make this paper more accessible to readers. In addition, with a nice initialization strategy, it converges to the global optimum under some assumptions.

This is a good paper. I was pleased to review this paper.

My general comments and suggestion are the following.

L177 : Authors claim that the method generalizes Wirtinger's flow algorithm which I'm not sure is right because the proposed method is defined with only real values (NOT complex); z and ai's. Please clarify that.

Please briefly explain about a "phase retrieval" for readers who are not familiar with the previous work by Candes et al [5].

Assumption on the true rank:

L183 : The paper assumes that the true rank is given prior to the algorithm. It might be a quite strong assumption in some sense and the nuclear norm method does not assume this. Discuss the effect of misspecification of rank with the proposed method.

L253 : As authors suggested, if one wants to search for the proper rank by running the algorithm multiple times, is there a systematic way of choosing r after running the algorithm say n number of times and looking at the value f(Z) after each r?

Eventually, the problem solved by the method is reduced to an optimization for a fixed (low) rank solution. In literature, it was addressed by multiple methods especially using manifold frameworks, which are not discussed in the paper. You may want to refer relevant works (R1-R3) for readers.

R1. Journee, Michel, et al. "Low-rank optimization on the cone of positive semidefinite matrices." SIAM Journal on Optimization 20.5 (2010): 2327-2351. R2. Vandereycken, Bart, and Stefan Vandewalle. "A Riemannian optimization approach for computing low-rank solutions of Lyapunov equations." SIAM Journal on Matrix Analysis and Applications 31.5 (2010): 2553-2579. R3. Vandereycken, Bart. "Low-rank matrix completion by Riemannian optimization."SIAM Journal on Optimization 23.2 (2013): 1214-1236.
Summary: This paper presents an efficient/scalable low-rank optimization algorithm on symmetric positive semidefinite matrices with linear convergence to the global optimum under some assumptions. I think this paper is a good paper and it would inspire further research on this topic.

Author Feedback
Author rebuttal: We thank the reviewers for their constructive comments, and respond briefly to the main comments and criticisms:

Reviewer 1
==========

1. The reviewer comments that the implementation may be far from state of the art. Please note that we have taken great care to make an accurate and fair comparison between the relevant methods discussed in the paper. In particular, Figure 2 shows a runtime versus accuracy comparison of SVP, gradient descent and nuclear norm. The implementations may not be the best possible, but they fairly compare the competing algorithms on moderately sized problems, supporting and illustrating the analysis. The codes and benchmarks such as Lenna mentioned by the reviewer do not appear to be directly relevant, as they target different problems.

2. The reviewer suggests that Section 6.1 about runtime may be misleading. We agree with the comment that complexity per iteration does not tell the whole story; thus the runtime comparison in Section 6.2 is more germane.

Reviewer 2
==========

3. We thank the reviewer for mentioning the papers of Burer and Monteiro (2005) and Lee and Bresler (2009). Both papers are certainly relevant related work, and should be discussed. The Burer and Monteiro (B&M) paper (with which we were previously familiar but neglected to cite) is important, and gives a helpful traceback of the factorization and nonconvex optimization idea in the optimization literature. While related, our algorithm and analysis are substantially different than these works. Essentially, B&M target the general semidefinite programming problem and have a more complex set of first order techniques for nonconvex optimization (BFGS and augmented Lagrangian techniques, etc.) It would not be easy to do a direct numerical comparison; but we would expect our methods to perform comparably. In contrast, our method is clean and simple, targets a more limited class of problems, and correspondingly allows us to obtain a strong theoretical convergence analysis (the pesky extra factor of r notwithstanding). As stated by Burer and Monteiro (2003) "Although we are able to derive some amount of theoretical justification for [global convergence], our belief that the method is not strongly affected by the inherent nonconvexity [of the objective function] is largely experimental." We hope that our work will contribute to and help spur the further development of this important class of techniques.

4. The sparsity parameter should be \rho rather than p. As the reviewer points out, the GOE corresponds to dense matrices. However, as shown in the empirical results, the computational advantages of our algorithm are magnified for sparse measurement matrices, which is why we include this parameter in Table 1.

Reviewer 3
==========

5. As the reviewer points out, the "Wirtinger flow" algorithm is for complex values in the phase retrieval setting. Our algorithm is thus not strictly speaking a generalization---we adapt the core idea of factoring the matrix variable and carrying out gradient descent on the factor.

6. Regarding the choice of rank, when the rank is incorrectly specified, our algorithm still quickly converges, but to a value that has relatively high error. When the correct rank is selected, the algorithm converges to a very low error. This results in a convenient, practical procedure for selecting the rank, as discussed. Further simulations can be included to clarify and reinforce this point.

7. We thank the reviewer for several relevant references. These papers present related techniques and should be included for completeness.

Reviewer 5
==========

8. The reviewer correctly points out that the analysis in the paper is suboptimal, with an extra factor of the rank r in the sample complexity. Our experiments strongly suggest that this extra factor is due to our currently loose analysis and is not intrinsic to the algorithm. As much as we would have liked to remove this factor, in our modest opinion, this does not lessen the potential impact of the paper. Rather, it simply points to a specific challenge for future work. A long and distinguished line of work in sparse data analysis (compressed sensing, lasso, etc.) has progressed through a series of refinements in sample complexity and assumptions.

9. The positive semidefinite constraint was dropped for nuclear norm minimization following the discussion in Section 2. If this constraint is imposed in the optimization the runtime of the nuclear norm method will degrade considerably.

Reviewer 6
==========

10. We respectfully point out that reference [i] mentioned by the reviewer is the very paper that our work is based on, and is discussed extensively. Paper [ii] addresses a different problem and uses a different method, though it could be listed in the previous work section.